# Initial Upper Palaeolithic humans in Europe had recent Neanderthal ancestry

Mateja Hajdinjak[1,2 ✉], Fabrizio Mafessoni[1], Laurits Skov[1], Benjamin Vernot[1], Alexander Hübner[1,3], Qiaomei Fu[4], Elena Essel[1], Sarah Nagel[1], Birgit Nickel[1], Julia Richter[1], Oana Teodora Moldovan[5,6], Silviu Constantin[7,8], Elena Endarova[9], Nikolay Zahariev[10], Rosen Spasov[10], Frido Welker[11,12], Geoff M. Smith[11], Virginie Sinet-Mathiot[11], Lindsey Paskulin[13], Helen Fewlass[11], Sahra Talamo[11,14], Zeljko Rezek[11,15], Svoboda Sirakova[16], Nikolay Sirakov[16], Shannon P. McPherron[11], Tsenka Tsanova[11], Jean-Jacques Hublin[11,17], Benjamin M. Peter[1], Matthias Meyer[1], Pontus Skoglund[2], Janet Kelso[1] & Svante Pääbo[1 ✉]

Modern humans appeared in Europe by at least 45,000 years ago[1–5], but the extent of their interactions with Neanderthals, who disappeared by about 40,000 years ago[6], and their relationship to the broader expansion of modern humans outside Africa are poorly understood. Here we present genome-wide data from three individuals dated to between 45,930 and 42,580 years ago from Bacho Kiro Cave, Bulgaria[1,2]. They are the earliest Late Pleistocene modern humans known to have been recovered in Europe so far, and were found in association with an Initial Upper Palaeolithic artefact assemblage. Unlike two previously studied individuals of similar ages from Romania[7] and Siberia[8] who did not contribute detectably to later populations, these individuals are more closely related to present-day and ancient populations in East Asia and the Americas than to later west Eurasian populations. This indicates that they belonged to a modern human migration into Europe that was not previously known from the genetic record, and provides evidence that there was at least some continuity between the earliest modern humans in Europe and later people in Eurasia. Moreover, we find that all three individuals had Neanderthal ancestors a few generations back in their family history, confirming that the first European modern humans mixed with Neanderthals and suggesting that such mixing could have been common.

The transition between the Middle and Upper Palaeolithic periods in Europe, which started about 47,000 years before present (47 kyr BP)[1,2], overlapped with the spread of modern humans and the disappearance of Neanderthals, which occurred by about 40 kyr BP[6]. Analyses of the genomes of Neanderthals and modern humans have shown that gene flow occurred between the two hominin groups approximately 60–50 kyr BP[8–11], probably in southwestern Asia. However, owing to the scarcity of modern human remains from Eurasia that are older than 40 kyr[1–5,12], genome-wide data are available for only three individuals of this age[7,8,13] (Fig. 1). Little is therefore known about the genetics of the earliest modern humans in Eurasia, the extent to which they interacted with archaic humans and their contribution to later populations. For example, whereas the roughly 42,000 to 37,000-year-old 'Oase1' individual from Romania[7,14] and the roughly 45,000-year-old 'Ust'Ishim' individual from Siberia[8] do not show specific genetic relationships to subsequent Eurasian populations, the approximately 40,000-year-old 'Tianyuan' individual from China contributed to the genetic ancestry of ancient and present-day East Asian populations[13]. Another open question is the extent to which modern humans mixed with Neanderthals when they spread across Europe and Asia. Direct evidence of local interbreeding exists only for the Oase1 individual, who had a recent Neanderthal ancestor[7] in his family history.

Here, we analyse genome-wide data from human specimens found in direct association with an Initial Upper Palaeolithic (IUP) assemblage of artefacts in Bacho Kiro Cave, Bulgaria[1] (Fig. 1), as well as from two more recent specimens from the same site (Supplementary Information 1). The IUP groups together assemblages that fall chronologically between the last Middle Palaeolithic assemblages and the first bladelet industries of the Upper Palaeolithic. The IUP spans a broad geographical area[15], from southwest Asia, central and eastern Europe to Mongolia[16] (Fig. 1,

[1]Department of Evolutionary Genetics, Max Planck Institute for Evolutionary Anthropology, Leipzig, Germany. [2]Francis Crick Institute, London, UK. [3]Department of Archaeogenetics, Max Planck Institute for the Science of Human History, Jena, Germany. [4]Key Laboratory of Vertebrate Evolution and Human Origins of Chinese Academy of Sciences, IVPP, Center for Excellence in Life and Paleoenvironment, Beijing, China. [5]Emil Racovita Institute of Speleology, Cluj Department, Cluj-Napoca, Romania. [6]Romanian Institute of Science and Technology, Cluj-Napoca, Romania. [7]Department of Geospeleology and Paleontology, Emil Racovita Institute of Speleology, Bucharest, Romania. [8]Centro Nacional de Investigación sobre la Evolución Humana, CENIEH, Burgos, Spain. [9]National History Museum, Sofia, Bulgaria. [10]Archaeology Department, New Bulgarian University, Sofia, Bulgaria. [11]Department of Human Evolution, Max Planck Institute for Evolutionary Anthropology, Leipzig, Germany. [12]Section for Evolutionary Genomics, GLOBE Institute, University of Copenhagen, Copenhagen, Denmark. [13]Department of Archaeology, University of Aberdeen, Aberdeen, UK. [14]Department of Chemistry 'G. Ciamician', University of Bologna, Bologna, Italy. [15]University of Pennsylvania Museum of Archaeology and Anthropology, University of Pennsylvania, Philadelphia, PA, USA. [16]National Institute of Archaeology with Museum, Bulgarian Academy of Sciences, Sofia, Bulgaria. [17]Chaire de Paléoanthropologie, Collège de France, Paris, France. ✉e-mail: mateja_hajdinjak@eva.mpg.de; paabo@eva.mpg.de

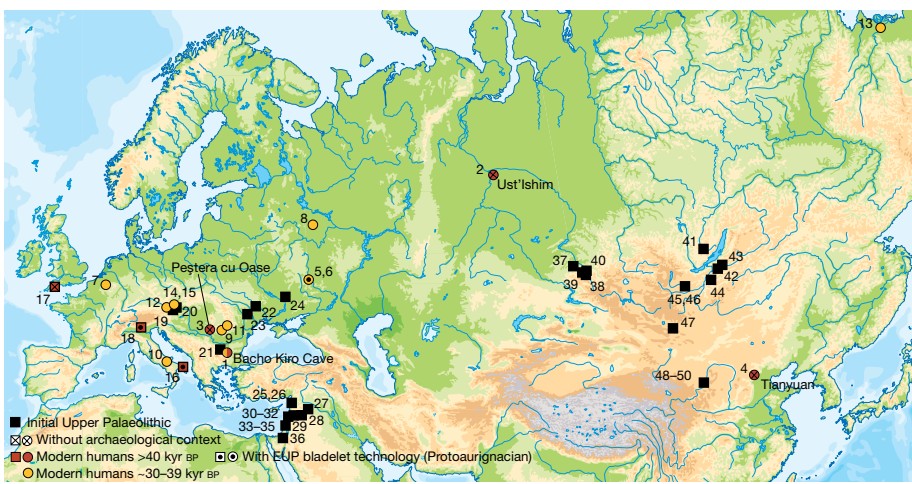

**Fig. 1 | Archaeological sites that have yielded genetic data and/or IUP assemblages.** Sites with modern human genome-wide data older than 40 kyr BP (red circles) or older than 30 kyr BP (yellow circles), sites in Europe with modern human remains older than 40 kyr BP (red squares) and sites with IUP assemblages (black squares).

1, Bacho Kiro Cave; 2, Ust'Ishim; 3, Peştera cu Oase; 4, Tianyuan Cave; 5, 6, Kostenki14 (Markina Gora) and Kostenki12 (Vokovskaya); 7, Troisème Caverne of Goyet; 8, Sunghir; 9, Peştera Muierilor; 10, Grotta Paglicci; 11, Peştera Cioclovina Uscată; 12, Krems Wachtberg; 13, Yana RHS; 14, 15, Dolní Věstonice and Pavlov; 16, Grotta del Cavallo; 17, Kents Cavern; 18, Grotta di Fumane; 19, Brno-Bohunice; 20, Stánska Skála III; 21, Temnata; 22, Kulychivka; 23, Korolevo 1 and 2; 24, Shlyakh; 25, 26, Üçagizli and Kanal Cave; 27, Um el'Tlel; 28, Jerf Ajlah; 29, Yabrud II; 30–32, Antelias; Abou Halka and Ksar Akil; 33–35, Emireh, El Wad and Raqefet; 36, Boker Tachtit; 37, Denisova Cave; 38, Kara-Bom; 39, Ust-Karakol 1; 40, Kara-Tenesh; 41, Makarvo IV; 42, Kamenka A–C; 43, Khotyk; 44, Podzvonkaya; 45, 46, Tolbor4 and Tolbor16; 47, Tsangan-Agui; 48–50, Suindonggou1, 2 and 9

Supplementary Information 1). Although there are reasons to group these assemblages on the basis of their lithic technology, the IUP also shows great regional variability. Therefore, it is debated whether the IUP represents a dispersal of modern humans across middle-latitude Eurasia, the diffusion of certain technological ideas, instances of independent invention, or a combination of some or all of these[15]. The IUP is contemporaneous with late Neanderthal sites in central and western Europe[6] and precedes later Upper Palaeolithic techno-complexes in Europe, such as the Protoaurignacian and the Aurignacian, by several thousand years[5].

Five human specimens were recovered from Bacho Kiro Cave in recent excavations. They consist of a lower molar (F6-620) found in the upper part of Layer J in the Main Sector, and four bone fragments (AA7-738, BB7-240, CC7-2289 and CC7-335) from Layer I in Niche 1. They have been directly radiocarbon-dated to between 45,930 and 42,580 calibrated years before present (cal. BP)[1,2], and their mitochondrial genomes are of the modern human type, suggesting that they are the oldest Upper Palaeolithic modern humans that have been recovered in Europe[1]. One bone fragment was found in Layer B in the Main Sector (F6-597) and another one was among the finds from excavations in the 1970s, when it was retrieved in a position corresponding to the interface of Layers B and C (BK1653). The two latter bone fragments were directly dated to 36,320–35,600 cal. BP and 35,290–34,610 cal. BP[1,2], respectively. Although the lithic assemblages from the later layers are sparse, they are likely to be Aurignacian[1,2]. We also produced additional data from a mandible[7,14] that was found outside any archaeological context in Peştera cu Oase, Romania (referred to as 'Oase1')[14]. The mandible was directly dated to about 42–37 kyr BP[14], although this may be an underestimate as the dating was performed before recent technical improvements.

We extracted DNA from between 29.3 mg and 64.7 mg of tooth or bone powder from the specimens as described[1]. We also treated 15 mg of bone powder from the Oase1 mandible with 0.5% hypochlorite solution to reduce bacterial and human contamination before DNA extraction[17]. Among DNA fragments sequenced from the DNA libraries constructed from the Bacho Kiro Cave and Oase1 extracts, between 0.003% and 1.8% could be mapped to the human genome (Supplementary Information 2). Owing to the low fraction of hominin DNA, we used in-solution hybridization capture[18] to enrich the libraries for about 3.8 million single-nucleotide polymorphisms (SNPs) that are informative about modern human variation and archaic admixture[7,19] (excluding F6-597, which contained very little if any endogenous DNA; Supplementary Information 2).

For the six specimens, between 57,293 and 3,272,827 of the targeted SNPs were covered by at least one DNA fragment (Extended Data Table 1). Of these, between 11,655 and 2,290,237 SNPs were covered by at least one fragment showing C-to-T substitutions in the first three and/or the last three positions from the ends, suggesting the presence of deaminated cytosine bases, which are typical of ancient DNA[20] (Extended Data Table 1, Extended Data Fig. 1). On the basis of the numbers of putatively deaminated fragments aligning to the X chromosome and the autosomes[21] (Supplementary Information 4), we conclude that specimens F6-620, AA7-738, BB7-240 and CC7-335 belonged to males, whereas BK1653 and CC7-2289 belonged to females, although the low amount of data makes this conclusion tentative for CC7-2289 (Extended Data Fig. 2a).

Using an approach that makes use of DNA deamination patterns[22], we estimated that the overall nuclear DNA contamination was between 2.2% ± 0.5% (F6-620) and 42.4% ± 0.6% (CC7-2289). In the male specimens, we estimated contamination from polymorphisms on the X chromosome[23] to between 1.6% ± 0.1% and 3.4% ± 0.5% (Supplementary Information 2). Owing to the presence of present-day human contamination, we restricted all downstream analyses to putatively deaminated fragments for all specimens except F6-620 (for which contamination was so low that we used all fragments). This left between 11,655 and 3,272,827 SNPs per specimen to be used for the subsequent analyses (Supplementary Information 2).

The molar F6-620 and the bone fragment AA7-738 have identical mitochondrial genome sequences[1] and both come from males. The pairwise mismatch rate between the two specimens at the SNPs[24] is 0.13, similar to the mismatch rate between libraries from the same specimen (Extended Data Fig. 2b). By contrast, this number is 0.23 (interquartile range: 0.22–0.25) for the other Bacho Kiro Cave specimens, similar to unrelated ancient individuals from other studies (Extended Data Fig. 2b). Thus, we conclude that specimens F6-620 and AA7-738 belonged to the same individual or to identical twins, which is much less likely.

We enriched the libraries from the male individuals using probes that targeted about 6.9 Mb of the Y chromosome[25] (Supplementary Information 4) and arrived at 15.2-fold coverage for F6-620, 2.5-fold for BB7-240 and 1.5-fold for CC7-335. F6-620 carries a basal lineage of the Y chromosome haplogroup F (F-M89), whereas BB7-240 and CC7-335 carry haplogroup C1 (C-F3393). Although haplogroup C is common among males from East Asia and Oceania, both haplogroups F and C1 are rare in present-day humans and are found only at low frequencies in mainland Southeast Asia and Japan[26,27].

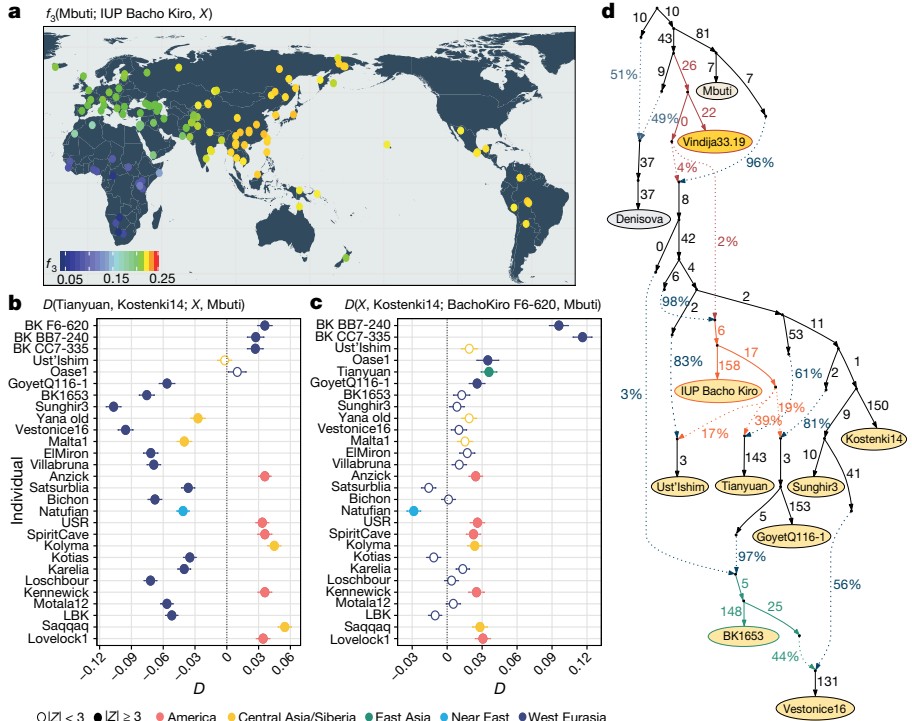

**Fig. 2 | Population affinities of the IUP Bacho Kiro Cave individuals. a**, Allele sharing ($f_3$) between the IUP Bacho Kiro Cave individuals and present-day populations ($X$) from the Simons Genome Diversity Project (SGDP)[31] after their separation from an outgroup (Mbuti) (calculated as $f_3$(Mbuti; IUP Bacho Kiro, $X$). Warmer colours on the map[48] correspond to higher $f_3$ values (higher shared genetic drift). **b**, IUP Bacho Kiro Cave individuals share significantly more alleles (proportions of allele sharing or $D$ values plotted on $x$ axis) with the roughly 40,000-year-old Tianyuan individual[13] than with the approximately 38,000-year-old Kostenki14 individual[29,30]. Calculated as $D$(Tianyuan, Kostenki14; $X$, Mbuti). **c**, F6-620 shares significantly more alleles with the Oase1[7] and GoyetQ116-1[29] individuals, ancient Siberians and Native American individuals than with the Kostenki14 individual. Calculated as $D$($X$, Kostenki14; F6-620, Mbuti). **b**, **c**, Filled circles indicate a significant value ($|Z| \geq 3$); open

circles, $|Z| < 3$. Whiskers correspond to 1 s.e. calculated across all autosomes (1,813,821 SNPs) using a weighted block jackknife[28] and a block size of 5 Mb. BK, Bacho Kiro. **d**, Admixture graph relating Bacho Kiro Cave individuals and ancient humans older than 30 kyr BP. This model uses 281,732 overlapping SNPs in all individuals and fits the data with a single outlier ($Z = 3.22$). Ancient non-Africans (yellow circles), Vindija 33.19 Neanderthal (orange), Denisovan (grey) and present-day African individuals (light yellow circle) are shown. Admixture edges (dotted lines) show the genetic component related to Neanderthals (red), to the IUP Bacho Kiro Cave individuals (orange) and to BK1653 (green). Numbers on solid branches correspond to the estimated drift in $f_2$ units of squared frequency difference; labels on dotted edges give admixture proportions.

We estimated the extent of genetic similarity among the Bacho Kiro Cave individuals and other early modern humans using outgroup $f_3$-statistics[28]. The three roughly 45,000-year-old IUP individuals are more similar to one another than to any other ancient individual (Extended Data Fig. 3a). By contrast, BK1653, which is about 35,000 years old, is more similar to later Upper Palaeolithic individuals from Europe who are around 38,000 years old or younger[29,30] ($3.0 \leq |Z| \leq 17.4$; Extended Data Fig. 4, Supplementary Information 5); for example, to the roughly 35,000-year-old 'GoyetQ116-1' individual from Belgium and members of the 'Věstonice' genetic cluster, who are associated with later Gravettian assemblages[29] (Extended Data Figs. 3a, 4b, c).

When comparing the Bacho Kiro Cave individuals to present-day populations[31], we found that the IUP individuals share more alleles (that is, more genetic variants) with present-day populations from East Asia, Central Asia and the Americas than with populations from western Eurasia (Fig. 2a, Supplementary Information 5), whereas the later BK1653 individual shares more alleles with present-day western Eurasian populations (Extended Data Figs. 3b, 4a).

We next investigated whether these observations could be due to the fact that present-day populations in western Eurasia derive part of their ancestry from 'Basal Eurasians'[32,33], an inferred population that diverged early from other non-African populations and may have 'diluted' allele sharing between western Eurasian populations and IUP individuals. To do this, we compared the Ust'Ishim, Oase1 and IUP Bacho Kiro Cave individuals to western Eurasian individuals such as the approximately

38,000-year-old 'Kostenki14' individual from Russia[29,30], which pre-dates the introduction of 'Basal Eurasian' ancestry to Europe around 8,000 cal. BP[32]. We found that the Ust'Ishim and Oase1 individuals showed no more affinity to western than to eastern Eurasian populations, suggesting that they did not contribute ancestry to later Eurasian populations, as previously shown[7,8] (Supplementary Information 5, Extended Data Fig. 5). By contrast, the IUP Bacho Kiro Cave individuals shared more alleles with the roughly 40,000-year-old Tianyuan individual[13] from China (Fig. 2b) and other ancient Siberians[34,35] and Native Americans[36–39] (Fig. 2c) than with the Kostenki14 individual ($3.6 \leq |Z| \leq 5.3$). Among other western Eurasian Upper Palaeolithic humans, the IUP Bacho Kiro Cave individuals shared more alleles with the Oase1 ($3.6 \leq |Z| \leq 4.3$) and roughly 35,000-year-old GoyetQ116-1[29] individuals than with the Kostenki14 individual ($3.2 \leq |Z| \leq 4.3$; Fig. 2c, Supplementary Information 5). Notably, the GoyetQ116-1 individual has previously been shown to share more alleles with early East Asians than other individuals of a similar age in Europe[13].

When we explored models of population history that are compatible with the observations above using admixture graphs[28], we found that the IUP Bacho Kiro Cave individuals were related to populations that contributed ancestry to the Tianyuan individual in China as well as, to a lesser extent, to the GoyetQ116-1 and Ust'Ishim individuals (all $|Z| < 3$; Fig. 2d, Supplementary Information 6). This resolves the previously unclear relationship between the GoyetQ116-1 and Tianyuan individuals[13] without the need for gene flow between these two geographically distant individuals. The models also suggest that the later BK1653

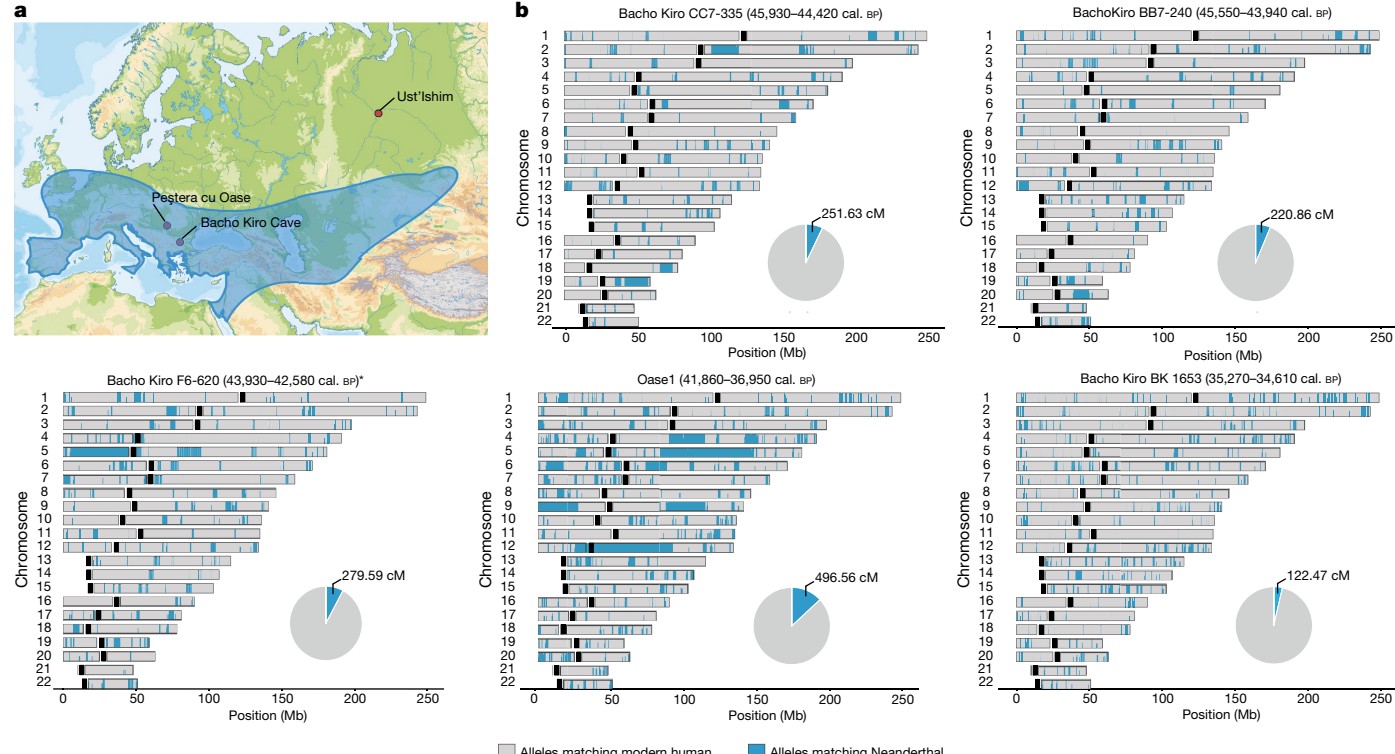

**Fig. 3 | Geographical distribution of Neanderthal archaeological sites and genome-wide distribution of Neanderthal alleles in the genomes of ancient modern humans. a**, Neanderthal geographical range (blue) and the locations of Peștera cu Oase, Bacho Kiro Cave and where the femur of the Ust'Ishim individual was found. **b**, Distribution of Neanderthal DNA in ancient modern human genomes. Neanderthal DNA segments longer than 0.2 cM are indicated in blue. Pie charts indicate the total proportion of Neanderthal DNA identified in each genome. Centromeres are shown in black.

individual belonged to a population that was related, but not identical, to that of the GoyetQ116-1 individual (Fig. 2d, Extended Data Fig. 4, Supplementary Information 6) and that the Věstonice cluster, whose members were found in association with Gravettian assemblages[29], derived part of their ancestry from such a population and the rest from populations related to the roughly 34,000-year-old 'Sunghir' individuals[40] from Russia (Fig. 2d, Supplementary Information 6).

As the IUP Bacho Kiro Cave individuals lived at the same time as some of the last Neanderthals in Europe[6], we estimated the proportion of Neanderthal DNA in their genomes by taking advantage of two high-quality Neanderthal genomes[9,10,41]. We found that the IUP individuals F6-620, BB7-240 and CC7-335 carried 3.8% (95% confidence interval (CI): 3.3–4.4%), 3.0% (95% CI: 2.4–3.6%) and 3.4% (95% CI: 2.8–4.0%) Neanderthal DNA, respectively. This is more than the average of 1.9% (95% CI: 1.5–2.4%) found in other ancient or present-day humans, except for the Oase1 individual, who had a close Neanderthal relative (6.4% (95% CI: 5.7–7.1%); Extended Data Fig. 6, Supplementary Information 7). By contrast, the more recent BK1653 individual carried only 1.9% (95% CI: 1.4–2.4%) Neanderthal DNA, similar to other ancient and present-day humans[10,41] (Extended Data Fig. 6). As has been the case for all humans studied so far, the Neanderthal DNA in BK1653 and the IUP Bacho Kiro Cave individuals was more similar to the Vindija33.19[10] and Chagyrskaya8[42] Neanderthals than to the Altai Neanderthal[9] ($2.8 \leq |Z| \leq 5.1$; Supplementary Information 7).

To study the spatial distribution of Neanderthal ancestry in the genomes of the Bacho Kiro Cave individuals, we used around 1.7 million SNPs at which Neanderthal[9] and/or Denisovan[43] genomes differ from African genomes[7] and an approach[44] that detects tracts of archaic DNA in ancient genomes. We found a total of 279.6 centiMorgans (cM) of Neanderthal DNA in F6-620, 251.6 cM in CC7-335 and 220.9 cM in BB7-240, and these individuals carried seven, six and nine Neanderthal DNA segments longer than 5 cM, respectively (Fig. 3, Extended Data Fig. 7a, Supplementary Information 8). The longest introgressed Neanderthal

segment in F6-620 encompassed 54.3 cM, and the longest segments in CC7-335 and BB7-240 were 25.6 cM and 17.4 cM, respectively (Fig. 3, Extended Data Fig. 7a). By contrast, the total amount of Neanderthal DNA in the BK1653 genome was 121.7 cM and the longest Neanderthal segment was 2.5 cM (Fig. 3, Extended Data Fig. 7a).

On the basis of the distribution of the long Neanderthal segments, we estimate that individual F6-620 had a Neanderthal ancestor less than six generations back in his family tree (Extended Data Table 2, Supplementary Information 8). Both the CC7-335 and BB7-240 individuals had Neanderthal ancestors about seven generations back in their families, with upper confidence intervals of ten and seventeen generations, respectively (Extended Data Table 2, Extended Data Fig. 7b, Supplementary Information 8). Thus, all IUP Bacho Kiro Cave individuals had recent Neanderthal ancestors in their immediate family histories.

To further explore the extent to which the Bacho Kiro Cave individuals contributed ancestry to later populations in Eurasia, we investigated whether the Neanderthal DNA segments in Bacho Kiro Cave genomes overlapped with Neanderthal segments in present-day populations more than expected by chance. We found more overlapping of segments between present-day East Asian populations and the IUP Bacho Kiro Cave individuals (lowest correlation coefficient of 0.09, 95% CI: 0.08–0.1) than with the BK1653 individual ($P = 0.02$, Wilcoxon test). By contrast, the BK1653 individual shows more overlapping of Neanderthal segments with present-day western Eurasian populations (a correlation coefficient of 0.11, 95% CI: 0.1–0.12) than do the IUP Bacho Kiro Cave individuals ($P < 1 \times 10^{-18}$, Wilcoxon test). This is compatible with the observation that the IUP Bacho Kiro Cave population contributed more ancestry to later populations in Asia and the Americas, whereas the BK1653 individual contributed more ancestry to populations in western Eurasia.

We next looked for overlap between parts of the human genome that carry little or no Neanderthal ancestry (Neanderthal 'deserts'), which are thought to have been caused by purifying selection against

Neanderthal DNA shortly after introgression[45,46]. We find almost no introgressed Neanderthal DNA in the previously described deserts in the IUP Bacho Kiro Cave and Oase1 individuals (249 kb out of 898 Mb of introgressed sequence; $P = 0.0079$, permutation $P$ value). When we restricted these comparisons to the more recent Neanderthal contributions (that is, segments longer than 5 cM), we similarly found no overlap (0 Mb out of 415 Mb, $P = 0.15$, permutation $P$ value), suggesting that selection against Neanderthal DNA variants occurred within a few generations, although additional individuals with recent Neanderthal ancestry will be needed to fully resolve this question.

In conclusion, the Bacho Kiro Cave genomes show that several distinct modern human populations existed during the early Upper Palaeolithic in Eurasia. Some of these populations, represented by the Oase1 and Ust'Ishim individuals, show no detectable affinities to later populations, whereas groups related to the IUP Bacho Kiro Cave individuals contributed to later populations with Asian ancestry as well as some western Eurasian humans such as the GoyetQ116-1 individual in Belgium. This is consistent with the fact that IUP archaeological assemblages are found from central and eastern Europe to present-day Mongolia[5,15,16] (Fig. 1), and a putative IUP dispersal that reached from eastern Europe to East Asia. Eventually populations related to the IUP Bacho Kiro Cave individuals disappeared in western Eurasia without leaving a detectable genetic contribution to later populations, as indicated by the fact that later individuals, including BK1653 at Bacho Kiro Cave, were closer to present-day European populations than to present-day Asian populations[29,30]. In Europe, the notion of successive population replacements is also consistent with the archaeological record, where the IUP is clearly intrusive against the Middle Palaeolithic background and where, apart from the common focus on blades, there are no clear technological connections between the IUP and the subsequent Aurignacian technologies. Finally, it is striking that all four of the European individuals who overlapped in time with late Neanderthals[7] and from whom genome-wide data have been retrieved had close Neanderthal relatives in their family histories (Fig. 3, Extended Data Figs. 7, 8). This suggests that mixing between Neanderthals and the first modern humans that arrived into Europe was perhaps more common than is often assumed.

*Note added in proof:* A companion paper[47] describes an individual from the Czechia who—based on genetic analyses—may be of similar age to the IUP Bacho Kiro Cave individuals and who carries a proportion of Neanderthal ancestry similar to later Upper Palaeolithic humans.

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

## Methods

### Ethics declaration

All approvals for specimen handling have been obtained from the relevant institutions. For the Oase1 specimen, the permission was granted to S.P. by the Emil Racovita Institute of Speleology, as the national authority in caves study. For the Bacho Kiro Cave specimens, the permission was granted by the Bulgarian Ministry of Culture and the National Museum of Natural History (Sofia, Bulgaria).

### DNA extraction and library preparation

Data generation for the seven Bacho Kiro Cave specimens (specimen IDs: F6-620, AA7-738, BB7-240, CC7-2289, CC7-335, F6-597 and BK1653) was based on DNA libraries prepared and described previously[1]. To obtain additional data from the Oase1 individual, we extracted DNA from 15 mg of bone powder from the specimen[7,14]. As it was previously found to be highly contaminated with microbial and present-day human DNA, the bone powder was treated with 0.5% hypochlorite solution before DNA extraction[17]. Four single-stranded DNA libraries were prepared from the resulting extract and two additional libraries were prepared, each using 5 μl of the two DNA extracts generated previously[7] as described[49]. The pools of libraries were then sequenced directly on Illumina MiSeq and HiSeq 2500 platforms in a double index configuration (2 × 76 cycles)[50] and base calling was done using Bustard (Illumina).

### DNA captures

We enriched the selected amplified libraries for about 3.7 million SNPs across the genome described in supplementary data 2 of ref. [19] (SNP Panel 1 or 390k array), and supplementary data 1–3 of ref. [7] (SNP Panels 2, 3 and 4, or 840k, 1000k and Archaic admixture arrays, respectively). For the male individuals (Bacho Kiro Cave F6-620, BB7-240 and CC7-335), an aliquot of each library was additionally enriched for about 6.9 Mb of the Y chromosome[25]. All of the enriched libraries were sequenced on the Illumina HiSeq 2500 platforms in a double index configuration (2 × 76 cycles)[50] and base calling was done using Bustard (Illumina).

### Sequencing of capture products and data processing

For all sequencing runs we trimmed the adapters and merged overlapping forward and reverse reads into single sequences using leeHom[51] (version: https://bioinf.eva.mpg.de/leehom/). The Burrows-Wheeler Aligner[52] (BWA, version: 0.5.10-evan.9-1-g44db244; https://github.com/mpieva/network-aware-bwa) with the parameters adjusted for ancient DNA ("-n 0.01 -o 2 -l16500")[43] was used to align the data from all sequencing runs to the human reference genome (GRCh37/1000 Genomes release; ftp://ftp.1000genomes.ebi.ac.uk/vol1/ftp/technical/reference/phase2_reference_assembly_sequence/). Only reads that showed perfect matches to the expected index combinations were used for all downstream analyses. PCR duplicates were removed using bam-rmdup (version: 0.6.3; https://github.com/mpieva/biohazard-tools) and SAMtools (version: 1.3.1)[53] was used to filter for fragments that were at least 35 bp long and that had a mapping quality equal to or greater than 25. BAM files of the libraries enriched for the specific subset of the nuclear genome were further intersected with the BED files containing target SNP positions (390k, 840k, 1000k, Archaic admixture, a merged set of SNP Panels 1 and 2 or 1240k, and a merged set of SNP Panels 1, 2 and 3 or 2200k) and regions (Y chromosome) using BEDtools[54] (version: 2.24.0). In order to filter for endogenous ancient DNA or putatively deaminated fragments, we used elevated C-to-T substitutions relative to the reference genome at the first three and/or last three positions of the alignment ends[20]. We merged libraries originating from the same specimen using samtools merge[53] to produce the final datasets for downstream analyses (Extended Data Table 1, Supplementary Information 2).

### Merging of the Bacho Kiro Cave and Oase1 data with other genomes

We performed random read sampling using bam-caller (https://github.com/bodkan/bam-caller, version: 0.1) by picking a base with a base quality of at least 30 at each position in the 1240k and 2200k SNP Panels that was covered by at least one fragment longer than 35 bp with a mapping quality equal to or higher than 25 ($L \geq 35$ bp, MQ $\geq 25$, BQ $\geq 30$). To mitigate the effect of deamination-derived substitutions on downstream analyses, we did not sample any Ts on the forward strands (in the orientation as sequenced) or any As on the reverse strands in the first five and/or last five positions from the alignment ends. Owing to the haploid nature of the Y chromosome, we called genotypes across the approximately 6.9 Mb of the Y chromosome for the enriched libraries of male individuals by calling a consensus allele at each position by majority call requiring a minimum coverage of 3 for specimens F6-620 and BB7-240 and of 2 for specimen CC7-335 using using bam-caller (https://github.com/bodkan/bam-caller, version: 0.1) (Supplementary Information 2).

We merged the data from the newly sequenced specimens with datasets of previously published ancient and present-day humans, as well as archaic humans, for three SNP panels (1240k, 2240k and Archaic admixture; Supplementary Information 3). Data from the 1240k panel include genotypes of 2,109 ancient and 2,974 present-day individuals compiled from published studies and available in the EIGENSTRAT format[28] at https://reich.hms.harvard.edu/allen-ancient-dna- resource-aadr-downloadable-genotypes-present-day-and-ancient-dna-data (version 37.2, released 22 February 2019). Data from the 2240k panel include published genetic data of ancient modern humans obtained through hybridization captures[7,13,29] and a range of present-day[9,31] and ancient modern humans[8,30,32–37,55–60], as well as the archaics[9,10,42,43,61], for which whole-genome shotgun data of varying coverage are available (Supplementary Information 3). The Archaic admixture panel data include 21 ancient modern humans directly enriched for these sites[7,13,29], as well as the genotypes of present-day[9,31] and ancient modern humans[8,30,32–37,55–60], as well as the archaics[9,10,42,43,61], for which whole-genome shotgun data are available (Supplementary Information 3) and that were intersected with about 1.7 million SNPs of the Archaic admixture panel using BEDTools[54] (version: 2.24.0).

### Population genetic analyses

To determine the relationship of the Bacho Kiro Cave and Oase1 individuals to other modern and archaic humans we used a range of $f$-statistics from ADMIXTOOLS[28] (version: v5.1) and as implemented in the R package admixr[62] (version: 0.7.1; Supplementary Information 4). We used qpGraph program (Admixture Graph) from ADMIXTOOLS[28] (version: v5.1) to test models of the relationship among Initial Upper Palaeolithic Bacho Kiro Cave individuals, the roughly 35,000-year-old Bacho Kiro Cave individual BK1653 and other ancient modern humans from Eurasia older than 30,000 years BP (Fig. 2d, Supplementary Information 6).

### Neanderthal ancestry

We estimated the proportion of Neanderthal DNA in the genomes of present-day and ancient modern humans by computing a direct $f_4$ ratio[28] that takes advantage of the two high-quality Neanderthal genomes[9,10,41] (Extended Data Fig. 6, Supplementary Information 7). We used admixfrog[44] (version: 0.5.6, https://github.com/BenjaminPeter/admixfrog/) to detect archaic introgressed segments in the genomes of the Bacho Kiro Cave and Oase1 individuals, as well as in other ancient modern humans and 254 present-day non-African individuals from the SGDP[31] as a direct comparison (Supplementary Information 8). We used these introgressed segments to estimate the number of generations since the most recent Neanderthal ancestor of the IUP Bacho Kiro Cave and Oase1 individuals (Supplementary Information 8), to investigate the overlap of Neanderthal segments in Bacho Kiro Cave individuals

with those detected in present-day and ancient modern humans (Supplementary Information 9), and to investigate the overlap of Neanderthal segments in the IUP Bacho Kiro Cave and Oase1 individuals with parts of the human genome devoid of Neanderthal ancestry[45,46] (Neanderthal deserts; Supplementary Information 10).

## Reporting summary

Further information on research design is available in the Nature Research Reporting Summary linked to this paper.

## Data availability

The aligned sequences of the Bacho Kiro Cave and Oase1 individuals have been deposited in the European Nucleotide Archive under accession number PRJEB39134. Comparative data of present-day human genomes from the SGDP that were used in this study are available at https://www.simonsfoundation.org/simons-genome-diversity-project/. Comparative data used in this study, which include genotypes of 2,109 ancient and 2,974 present-day individuals compiled from published studies, are available in the EIGENSTRAT file format at https://reich.hms.harvard.edu/allen-ancient-dna-resource-aadr-downloadable-genotypes-present-day-and-ancient-dna-data (version 37.2, released 22 February 2019). To determine the Y chromosome haplogroups of male individuals in this study, we used the Y-haplogroup tree from the International Society of Genetic Genealogy (ISOGG, available at http://www.isogg.org, version: 13.38).

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

**Acknowledgements** We thank A. Weihmann and B. Schellbach for their help with DNA sequencing; R. Barr, P. Korlević and S. Tüpke for help with graphics; D. Reich and M. Slatkin for discussions and input; the Tourist Association STD "Bacho Kiro" in Dryanovo; the History Museum in Dryanovo; the Regional History Museum in Gabrovo; the National Museum of Natural History (NMNH) in Sofia; and N. Spassov. M.H. is supported by a Marie Skłodowska Curie Individual Fellowship (no. 844014). Q.F. was supported by the Strategic Priority Research Program (B) (XDB26000000) of CAS, NSFC (41925009, 41630102, 41672021). O.T.M. and S.C. were supported by a grant from the Ministry of Research and Innovation, CNCS - UEFISCDI, project number PN-III-P4-ID-PCCF-2016-0016, within PNCDI III and the EEA Grants 2014-2021, under Project contract no. 3/2019. F.W. received funding from the European Research Council (ERC) under the European Union's Horizon 2020 research and innovation programme (grant agreement no. 948365). P.S. was supported by the Vallee Foundation, the European Research Council (grant no. 852558), the Wellcome Trust (217223/Z/19/Z) and Francis Crick Institute core funding (FC001595) from Cancer Research UK, the UK Medical Research Council and the Wellcome Trust. This study was funded by the Max Planck Society and the European Research Council (grant agreement no. 694707 to S.P.).

**Author contributions** M.H., E. Essel, S.N., B.N., J.R. and Q.F. performed ancient DNA lab work. O.T.M., S.C., E. Endarova, N.Z., R.S., F.W., G.M.S., V.S.-M., H.F., S.T., Z.R., S.S., N.S., S.P.M., T.T. and J.-J.H. provided and analysed archaeological material. M.H., F.M., L.S., B.V., A.H. and B.M.P. analysed DNA data. B.M.P., M.M., P.S., J.K. and S.P. supervised the study. M.H., J.K. and S.P. wrote the manuscript with input from all co-authors.

**Funding** Open access funding provided by Max Planck Society.

**Competing interests** The authors declare no competing interests.

**Additional information**
**Correspondence and requests for materials** should be addressed to M.H. or S.P.

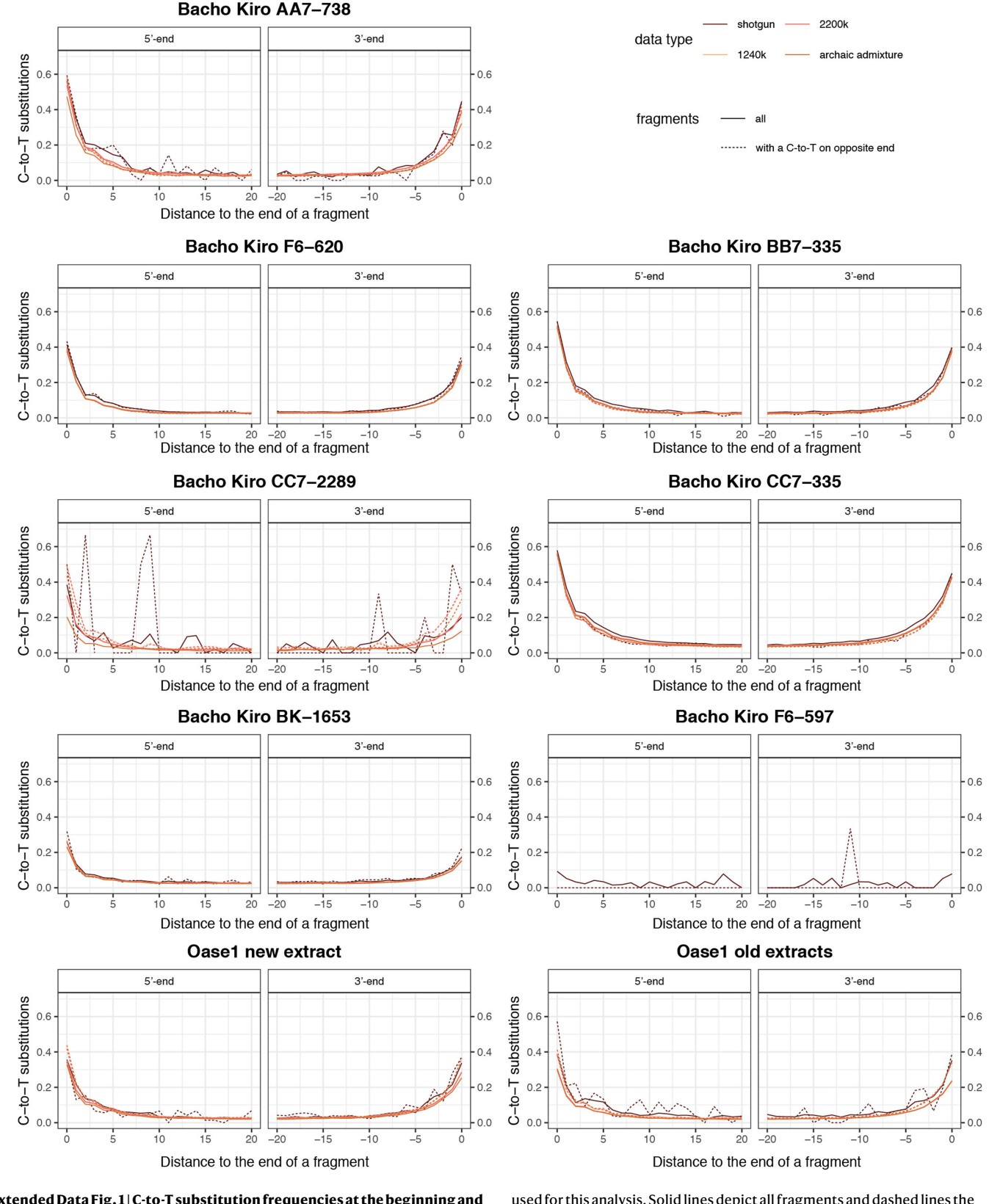

**Extended Data Fig. 1 | C-to-T substitution frequencies at the beginning and end of nuclear alignments for the merged libraries of the Bacho Kiro Cave and Oase1 specimens.** Only fragments of at least 35 bp that mapped to the human reference genome with a mapping quality of at least 25 (MQ ≥ 25) were used for this analysis. Solid lines depict all fragments and dashed lines the fragments that have a C-to-T substitution at the opposing end (conditional substitutions).

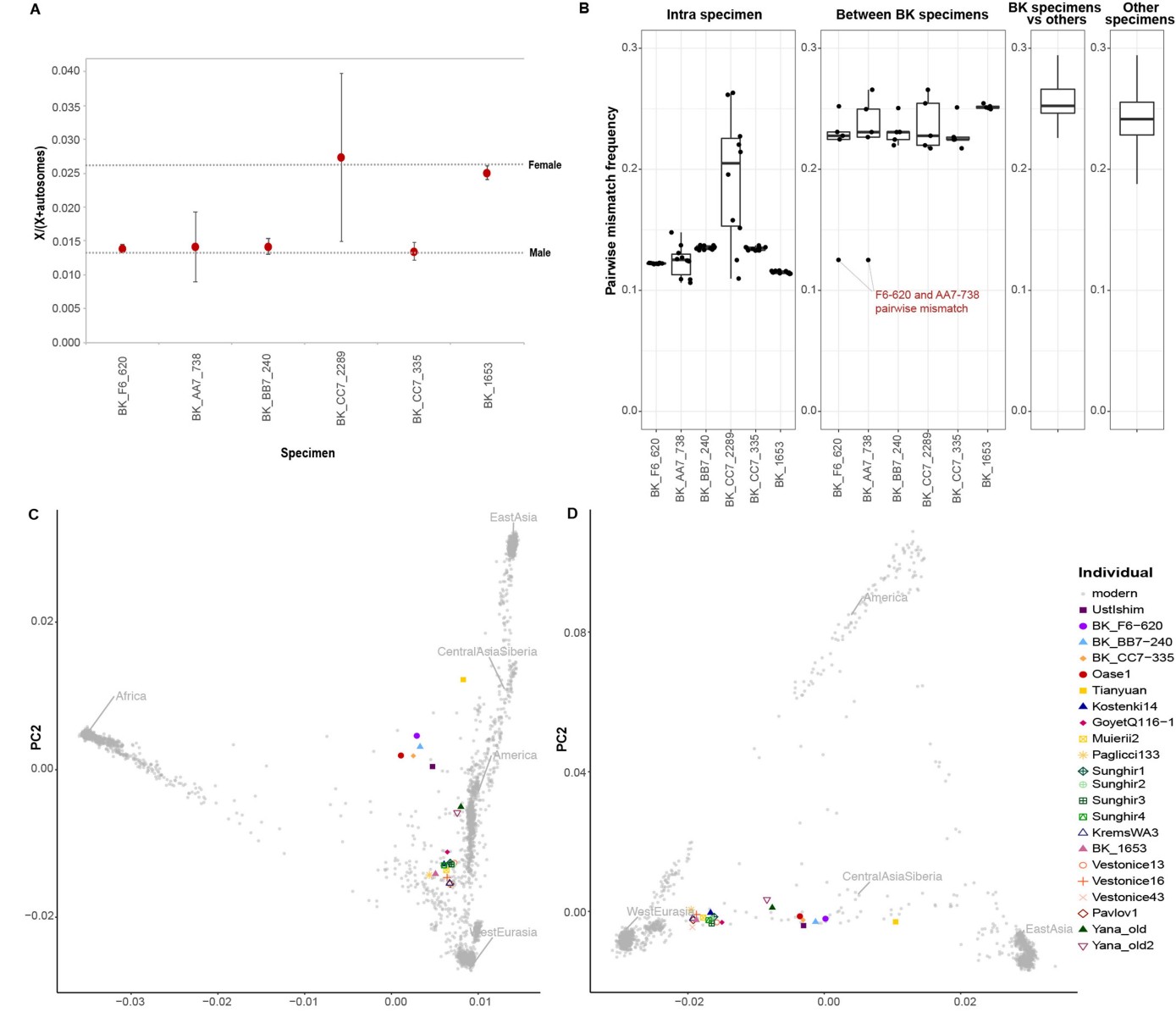

**Extended Data Fig. 2 | Sex determination, pairwise mismatch rate between specimens and principal component analyses (PCAs). a**, Sex determination for Bacho Kiro Cave specimens. Only fragments that showed C-to-T substitutions in the first three and/or last three positions and overlapping 2200k Panel SNPs were used for this analysis (for the number of deaminated fragments per specimen, see Supplementary Table 2.8). The expected ratios of X to (X + autosomal) fragments for a female and a male individual are depicted as dashed lines, and circles correspond to the calculated values for each of the Bacho Kiro Cave specimens. Whiskers correspond to 95% binomial confidence intervals. **b**, Pairwise mismatch rate between different libraries from the same specimen (intra-specimen), between different Bacho Kiro Cave specimens (inter-specimen) and between other ancient modern humans older than 30,000 cal. ʙᴘ. The boxplots were drawn using the summary statistics geom_stat from the R-package ggplot; lower and upper hinges, first and third quartiles; whiskers, maximum value of 1.5× the interquartile range; centre line, median. SNPs across all autosomes of the 2200k Panel were used for the calculations (number of SNPs (nsnps) = 2,056,573). **c**, A PCA of 2,970 present-day humans genotyped on 597,573 SNPs with 22 ancient individuals older than 30,000 cal. ʙᴘ projected onto the plane. **d**, A PCA of 1,444 present-day Eurasian and Native American individuals genotyped on 597,573 SNPs with 22 ancient individuals older than 30,000 cal. ʙᴘ projected onto the plane. **c**, **d**, Grey dots denote present-day human genomes.

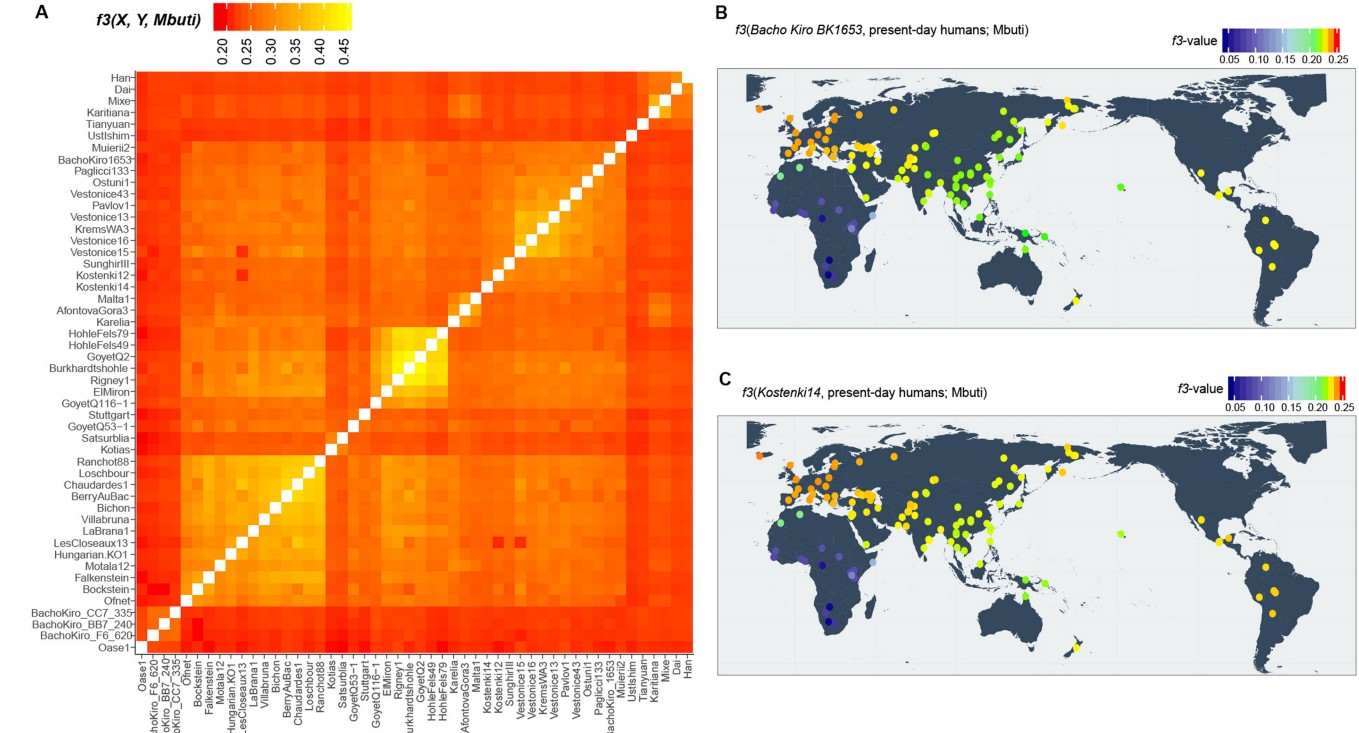

**Extended Data Fig. 3 | Heatmaps of outgroup $f_3$-statistics corresponding to the amount of shared genetic drift between individuals and/or populations. a**, Genetic clustering of ancient individuals, including the IUP Bacho Kiro Cave, BK1653 and Oase1 individuals based on the amount of shared genetic drift and calculated as $f_3$(ancient$_1$, ancient$_2$; Mbuti). Lighter colours in this panel indicate higher $f_3$ values and correspond to higher shared genetic drift (nsnps = 2,056,573). **b, c**, Shared genetic drift between the approximately 35,000-year-old BK1653 (**b**; nsnps = 825,379) or approximately 38,000-year-old

Kostenki14 individuals[29,30] (**c**; nsnps = 1,676,430) and present-day human populations from the SGDP[31] calculated as $f_3$(Bacho Kiro BK1653/Kostenki14, present-day humans; Mbuti). Three Mbuti individuals from the same panel[31] were used as an outgroup. Higher $f_3$ values[47] are indicated with warmer colours and correspond to higher shared genetic drift. Plotted $f_3$ values were calculated using ADMIXTOOLS[28] as implemented in admixr[61]. Coordinates for present-day humans were previously published[31]. The heatmap scale is consistent with those in Fig. 2a, Supplementary Figs. 5.1, 5.2.

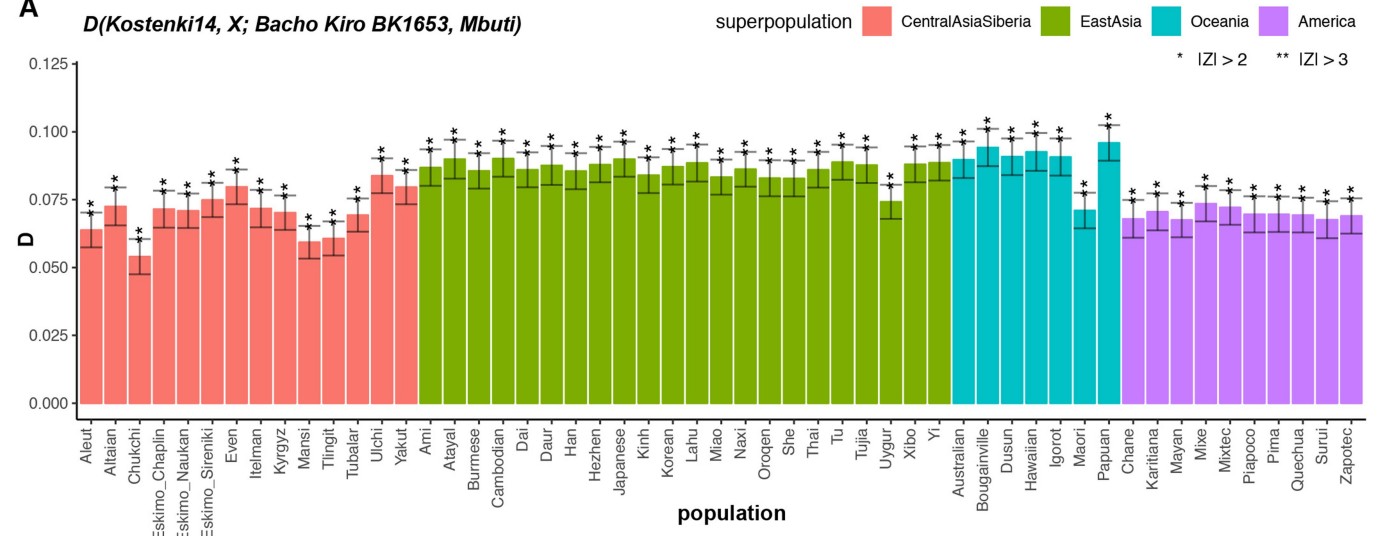

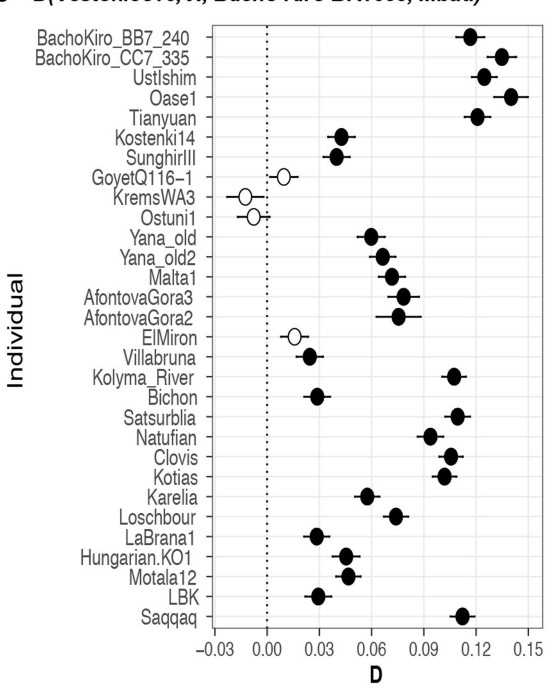

**Extended Data Fig. 4 | Population affinities of the approximately 35,000-year-old BK1653 individual. a,** In contrast to the IUP Bacho Kiro Cave individuals, individual BK1653 is significantly closer to the approximately 38,000-year-old Kostenki14 individual[29,30] than to present-day non-African populations from Central Asia and Siberia, East Asia, South Asia, Oceania or the Americas, as calculated by $D$(Kostenki14, present-day humans; BachoKiro BK1653, Mbuti). $D$ values for each comparison, plotted as barplots, were calculated using ADMIXTOOLS[28] as implemented in admixr[61]. Present-day human genomes from the SGDP[31] were used in these statistics, and three Mbuti

individuals from the same panel were used as an outgroup. **|Z| ≥ 3, *|Z| ≥ 2. **b, c,** BK1653 shares significantly more alleles with the approximately 35,000-year-old GoyetQ116-1[29] (**b**) and approximately 31,000-year-old Vestonice16[29] individuals (**c**) than with most other ancient modern humans. D values calculated as in **a**. Filled circles correspond to |Z| ≥ 3, and open circles indicate a |Z| < 3 (not significant). Error bars in all panels show s.e. calculated using a weighted block jackknife[28] across all autosomes on the 2200k Panel (nsnps (Bacho Kiro BK1653) = 825,379) and a block size of 5 Mb.

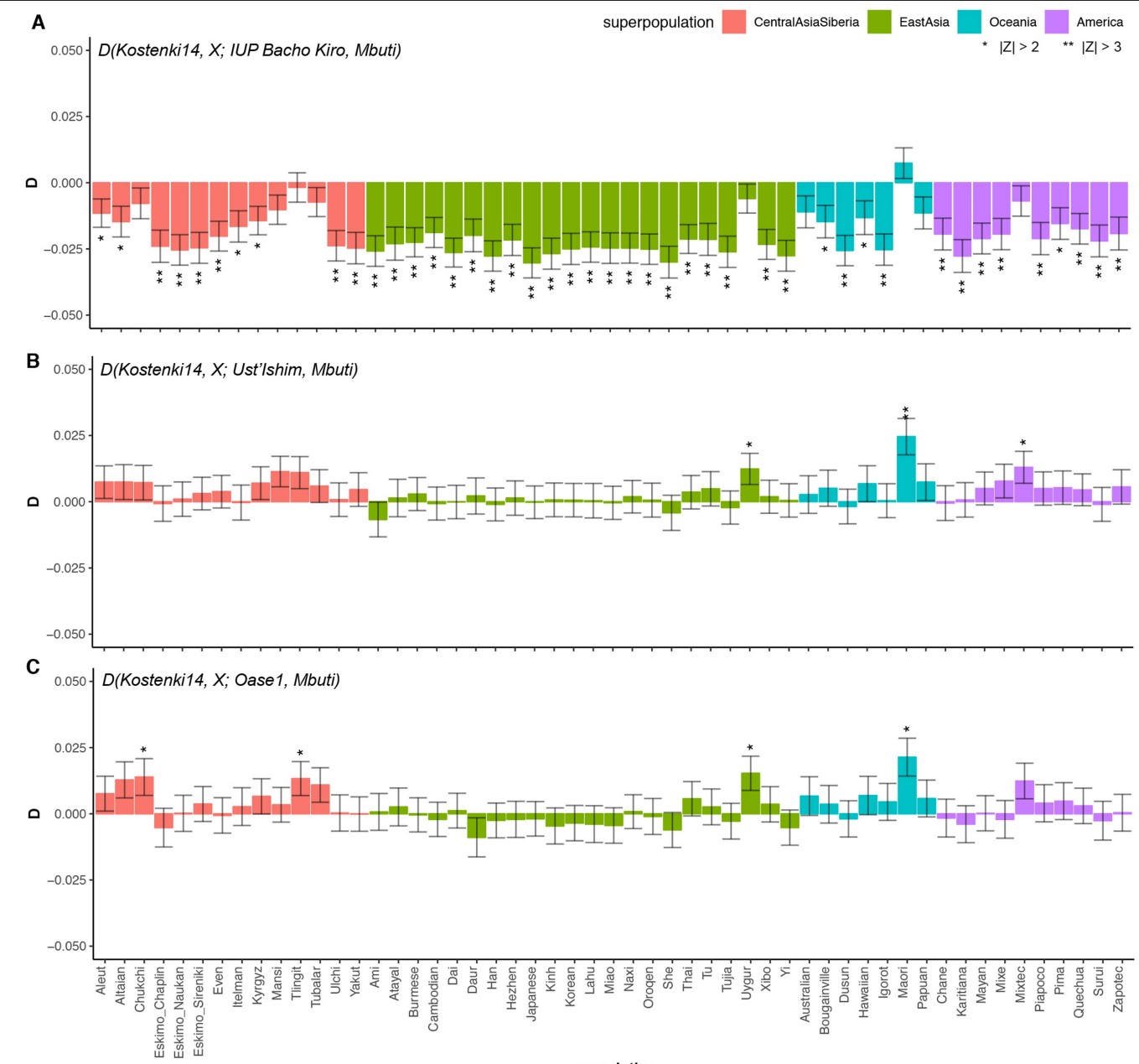

**Extended Data Fig. 5 | *D*(Kostenki14, *X*; IUP Bacho Kiro Cave individuals/ Ust'Ishim/Oase1, Mbuti) where *X* is a present-day non-African population from Central Asia and Siberia, East Asia, South Asia, Oceania or the Americas.** *D* values for each comparison, plotted as barplots, were calculated using ADMIXTOOLS[28] as implemented in admixr[61]. Present-day human genomes from the SGDP[31] were used in these statistics, and three Mbuti individuals from the same panel were used as an outgroup. \*\*|*Z*| ≥ 3, \*|*Z*| ≥ 2. Error bars denote s.e. calculated using a weighted block jackknife[28] across all autosomes on the 2200k Panel and a block size of 5 Mb. **a**, A pool of three IUP Bacho Kiro Cave individuals (nsnps = 1,813,821). **b**, Ust'Ishim (nsnps = 1,951,462). **c**, Oase1 (nsnps = 402,526).

**A**

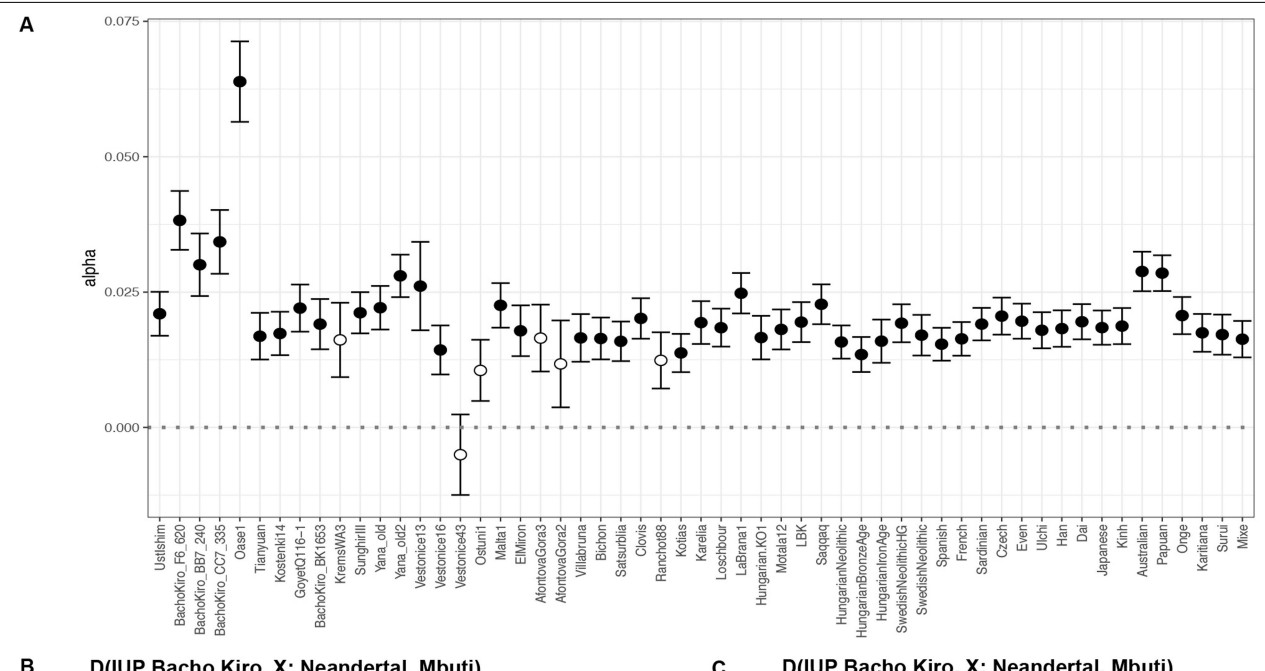

**B**  D(IUP Bacho Kiro, X; Neandertal, Mbuti)

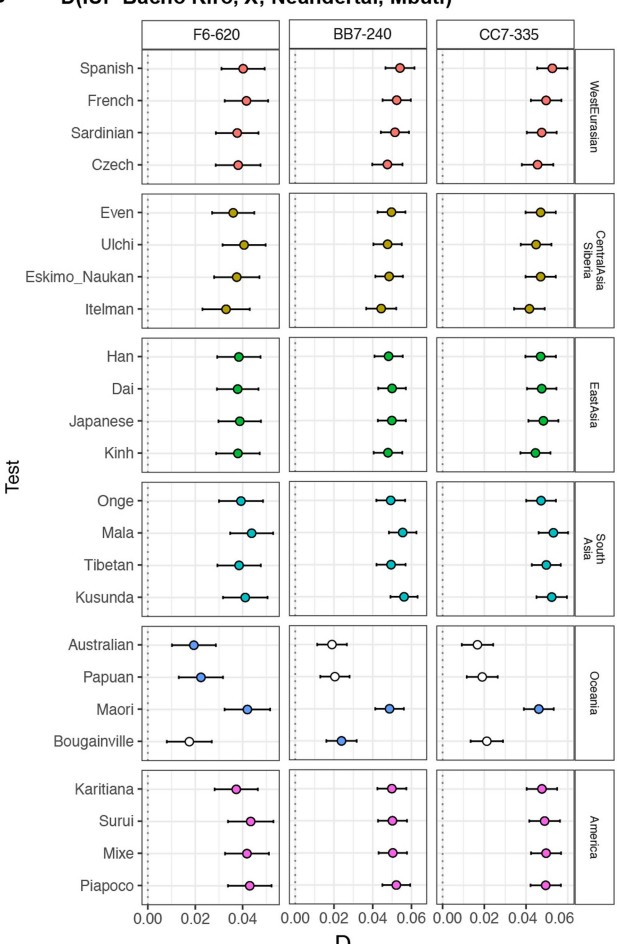

**C**  D(IUP Bacho Kiro, X; Neandertal, Mbuti)

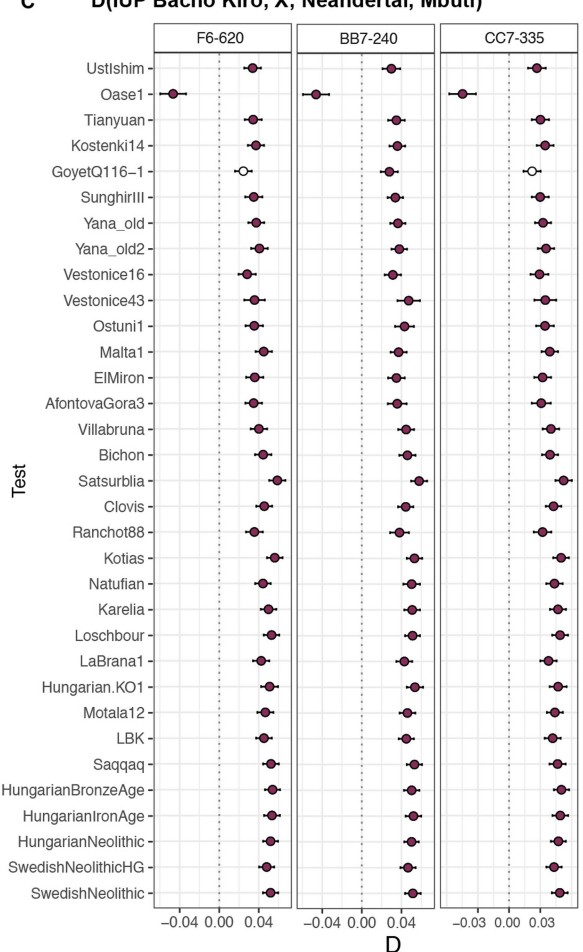

**Extended Data Fig. 6 | Neanderthal ancestry in IUP Bacho Kiro Cave individuals. a**, The proportion of Neanderthal ancestry in Bacho Kiro Cave individuals and other ancient and present-day modern humans calculated with a direct $f_4$ ratio that takes advantage of the two high-coverage Neanderthal genomes[9,10,41]. $f_4$ ratio (alpha) values calculated using ADMIXTOOLS[28] as implemented in admixr[61]. **b, c**, Neanderthals[9,10,42] share significantly more derived alleles with the IUP Bacho Kiro Cave individuals than with most present-day[31] (**b**) or ancient modern humans (**c**). $D$ values calculated using ADMIXTOOLS[28] as implemented in admixr[61]. Filled circles correspond to $|Z| \geq 3$; open circles indicate $|Z| < 3$ (not significant). Error bars in all panels show s.e. calculated using a weighted block jackknife[28] across all autosomes on the 2200k Panel (nsnps = 2,056,573) and a block size of 5 Mb.

**A** Genomic positions of archaic fragments
Using fragments longer than 0.2 cM
Light grey areas are deserts > 10 Mb and grey are deserts > 1 Mb. Black regions are gaps > 1 Mb in hg19

**Individual**
- BachoKiro_AA7_738 (same individual as F6-620)
- BachoKiro_F6-620
- BachoKiro_CC7-2289
- BachoKiro_BB7-240
- BachoKiro_CC7-335
- BachoKiro_BK1653
- Oase1

**B**

**Extended Data Fig. 7 | Segments of Neanderthal ancestry and estimates of the number of generations since the most recent Neanderthal ancestor. a**, A combined plot of the inferred Neanderthal segments in the genomes of the IUP Bacho Kiro Cave, BK1653 and Oase1 individuals, including chromosome X, using a hidden Markov model approach (admixfrog)[44]. **b**, Maximum likelihood estimates (dashed red lines) of the number of generations since a recent additional Neanderthal introgression into the IUP Bacho Kiro Cave and Oase1 individuals. Dashed black lines show 95% confidence intervals.

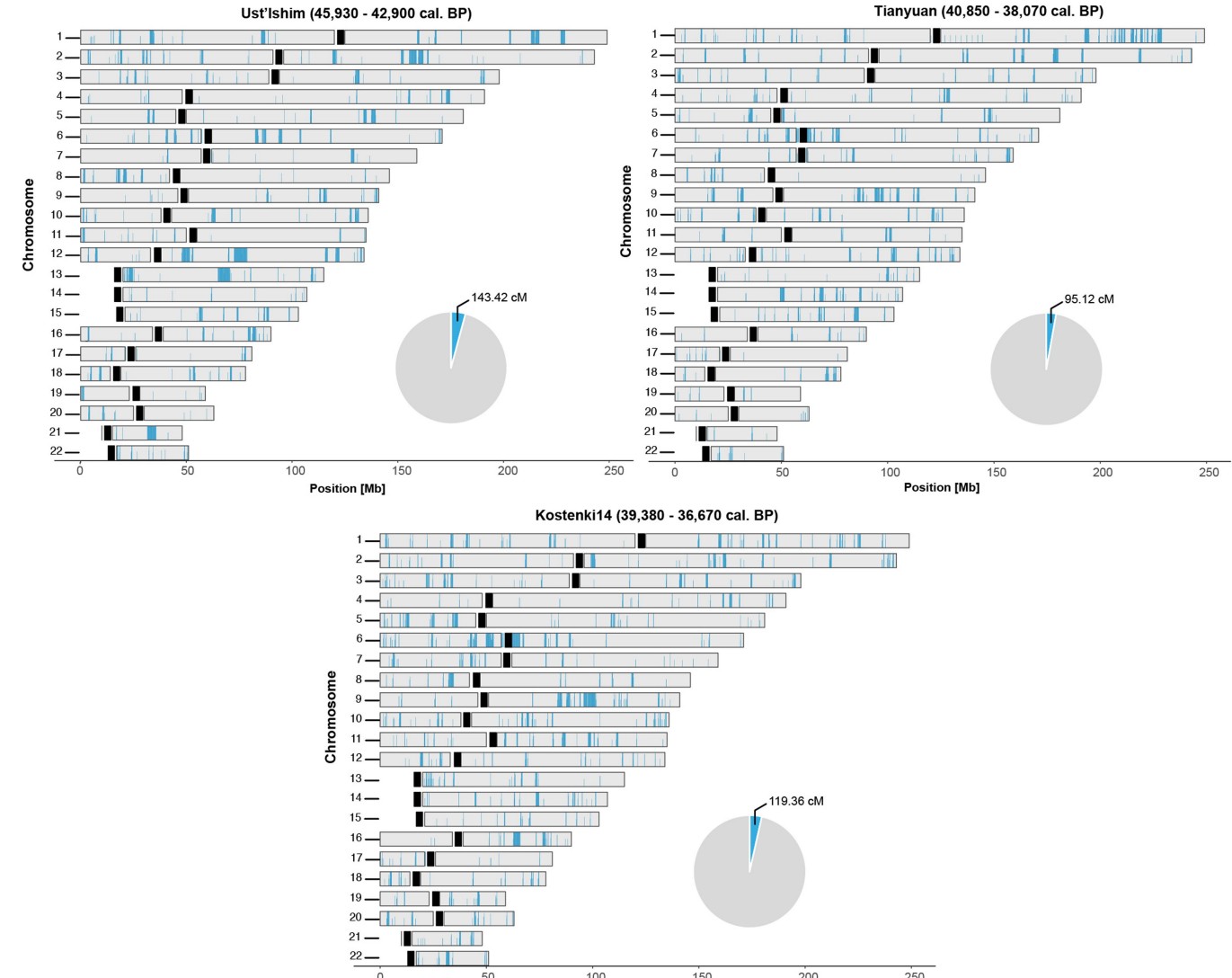

**Extended Data Fig. 8 | Spatial distribution of Neanderthal DNA in the Ust'Ishim, Tianyuan and Kostenki14 genomes.** Segments corresponding to Neanderthal ancestry inferred using a hidden Markov model approach (admixfrog)[44] longer than 0.2 cM are indicated in blue. Centromeres are indicated in black. Pie charts indicate the total amount of Neanderthal DNA identified in each genome.

## Extended Data Table 1 | Amount of data generated for Bacho Kiro Cave and Oase1 libraries for each SNP panel

| SNP panel | Specimen | Number of sequenced fragments | Number of fragments ≥35 bp | Number of mapped fragments ≥35bp, MQ≥25 | Number of mapped fragments on target ≥35bp, MQ≥25 | Number of unique fragments on target ≥35bp, MQ≥25 | Number of SNPs on target | % of SNPs on target | Number of deaminated fragments on target ≥35bp, MQ≥25 | Number of SNPs on target | % of SNPs on target |
|---|---|---|---|---|---|---|---|---|---|---|---|
| 390k | F6-620 | 42,944,497 | 31,607,676 | 10,664,089 | 5,417,563 | 3,610,135 | 372,571 | 94.61 | 879,932 | 294,287 | 74.73 |
| | AA7-738 | 43,905,093 | 22,343,746 | 579,771 | 68,137 | 29,434 | 26,299 | 6.68 | 9,975 | 9,629 | 2.45 |
| | BB7-240 | 48,679,210 | 32,531,469 | 5,428,952 | 1,666,056 | 893,179 | 311,587 | 79.13 | 307,661 | 191,684 | 48.68 |
| | CC7-2289 | 16,999,931 | 12,126,017 | 51,649 | 19,810 | 9,097 | 8,659 | 2.2 | 1,828 | 1,816 | 0.46 |
| | CC7-335 | 46,334,184 | 33,761,817 | 6,255,866 | 1,557,930 | 707,423 | 290,832 | 73.85 | 263,187 | 174,782 | 44.38 |
| | BK1653 | 68,575,246 | 42,484,607 | 13,844,793 | 4,326,253 | 2,510,542 | 393,788 | 93.41 | 381,268 | 215,802 | 54.8 |
| | Oase1 | 59,102,089 | 38,286,426 | 1,846,880 | 1,320,207 | 223,401 | 155,508 | 39.49 | 48,556 | 44,591 | 11.32 |
| 840k | F6-620 | 68,861,832 | 52,019,777 | 19,021,767 | 13,169,868 | 6,350,393 | 757,051 | 89.84 | 1,560,036 | 558,143 | 66.24 |
| | AA7-738 | 62,671,011 | 46,507,342 | 790,242 | 478,993 | 64,038 | 58,037 | 6.89 | 21,929 | 21,376 | 2.54 |
| | BB7-240 | 72,044,536 | 49,699,585 | 9,815,710 | 6,412,040 | 1,650,434 | 610,752 | 72.48 | 571,213 | 362,810 | 43.06 |
| | CC7-2289 | 58,615,440 | 42,560,335 | 150,916 | 71,215 | 18,721 | 18,151 | 2.15 | 3,716 | 3,731 | 0.44 |
| | CC7-335 | 68,794,275 | 51,231,565 | 10,390,207 | 6,599,859 | 1,290,666 | 561,737 | 66.66 | 479,284 | 326,926 | 38.8 |
| | BK1653 | 66,744,975 | 49,155,219 | 14,281,786 | 9,230,491 | 3,840,020 | 694,592 | 82.43 | 594,397 | 356,855 | 43.35 |
| | Oase1 | 56,745,317 | 37,030,590 | 3,031,789 | 1,920,548 | 368,252 | 270,601 | 32.11 | 80,709 | 75,569 | 8.97 |
| 1000k | F6-620 | 69,608,818 | 53,289,756 | 19,230,002 | 13,384,508 | 5,313,451 | 808,722 | 81.05 | 1,255,469 | 537,525 | 53.87 |
| | AA7-738 | 61,551,239 | 42,095,902 | 677,982 | 398,801 | 41,360 | 39,660 | 3.97 | 13,567 | 13,677 | 1.37 |
| | BB7-240 | 75,390,565 | 51,844,259 | 8,990,852 | 5,821,596 | 1,209,082 | 565,852 | 56.71 | 402,866 | 294,403 | 29.51 |
| | CC7-2289 | 54,944,426 | 40,633,321 | 167,548 | 64,293 | 9,745 | 10,003 | 1 | 1,874 | 1,965 | 0.2 |
| | CC7-335 | 72,464,739 | 54,221,955 | 10,907,573 | 6,935,567 | 1,062,963 | 536,521 | 53.77 | 383,734 | 286,514 | 28.72 |
| | BK1653 | 72,387,295 | 49,998,398 | 14,529,900 | 9,637,535 | 2,911,357 | 726,130 | 72.77 | 434,351 | 305,619 | 30.63 |
| | Oase1 | 59,348,519 | 38,586,583 | 2,347,235 | 1,434,618 | 272,943 | 221,020 | 22.15 | 56,420 | 55,309 | 5.54 |
| Archaic admixture | F6-620 | 153,782,894 | 113,385,420 | 31,863,327 | 20,457,629 | 8,262,261 | 1,405,078 | 80.32 | 2,004,756 | 927,570 | 53.02 |
| | AA7-738 | 74,474,294 | 55,404,191 | 741,606 | 316,366 | 72,962 | 67,495 | 3.86 | 22,510 | 24,423 | 1.4 |
| | BB7-240 | 159,324,081 | 104,745,219 | 11,377,514 | 6,359,192 | 1,852,195 | 947,950 | 54.19 | 632,972 | 500,142 | 28.59 |
| | CC7-2289 | 77,644,299 | 56,438,550 | 351,596 | 78,994 | 28,065 | 19,703 | 1.13 | 4,048 | 3,959 | 0.27 |
| | CC7-335 | 156,715,471 | 112,448,992 | 14,570,648 | 8,134,133 | 1,650,897 | 907,904 | 51.9 | 606,177 | 488,615 | 27.93 |
| | BK1653 | 159,168,237 | 111,693,225 | 22,866,297 | 13,162,057 | 4,591,355 | 1,257,365 | 71.87 | 702,281 | 533,274 | 30.48 |
| | Oase1 | 52,319,743 | 33,767,952 | 1,719,613 | 890,910 | 306,637 | 278,234 | 15.9 | 64,447 | 71,112 | 4.06 |
| "2200k" | F6-620 | 181,415,147 | 136,917,209 | 48,915,858 | 35,088,657 | 15,022,650 | 1,867,749 | 87.09 | 3,639,282 | 1,362,667 | 64.54 |
| | AA7-738 | 168,127,343 | 110,946,990 | 2,047,995 | 1,328,039 | 131,193 | 123,265 | 5.75 | 44,375 | 44,610 | 2.08 |
| | BB7-240 | 196,114,311 | 134,075,313 | 24,235,514 | 16,377,366 | 3,662,866 | 1,451,175 | 67.67 | 1,252,736 | 840,421 | 39.19 |
| | CC7-2289 | 130,559,797 | 95,319,673 | 370,113 | 169,597 | 37,238 | 37,590 | 1.75 | 7,372 | 7,696 | 0.36 |
| | CC7-335 | 187,593,198 | 139,215,337 | 27,553,646 | 18,181,311 | 2,967,786 | 1,353,413 | 63.11 | 1,093,477 | 776,283 | 36.2 |
| | BK1653 | 207,707,516 | 141,638,224 | 42,656,479 | 29,863,646 | 9,098,112 | 1,728,159 | 80.59 | 1,387,333 | 874,287 | 40.77 |
| | Oase1 | 175,195,925 | 113,903,599 | 7,225,904 | 4,722,435 | 850,586 | 646,646 | 30.15 | 183,143 | 177,336 | 8.27 |

Table shows number of fragments after merging all of the sequencing libraries together for each specimen. Fragments longer than 35 bp with a mapping quality of at least 25 that overlapped different target sites were used for reporting the number of SNPs on target.

**Extended Data Table 2 | Estimates of the number of generations before the most recent Neanderthal introgression in Bacho Kiro Cave and Oase1 individuals (tracts >5 cM) obtained by calculating the complementary cumulative distribution (CCD) of the lengths of Neanderthal tracts**

| Individual | tracts > 5cM<br>n.generations (+/- 95% CI) | tracts ≤ 5cM<br>n.generations (+/- 95% CI) |
|---|---|---|
| BachoKiro BB7-240 | 11.9 (6.9-17.0) | 89.9 (87.9-92.0) |
| BachoKiro CC7-2289 | 7.7 (3.4-11.9) | 60.3 (50.7- 70.0) |
| BachoKiro CC7-335 | 7.4 (5.0 -9.8) | 99.3 (95.7- 102.6) |
| BachoKiro F6-620 | 3.9 (1.9-6.0) | 87.7 (85.8- 89.7) |
| Oase1 | 3.9 (3.4-4.6) | 89.0 (86.6-91.2) |

Estimates for tracts ≤5 cM and therefore representative of older introgression events are also shown.

|---|---|

# Reporting Summary

Nature Research wishes to improve the reproducibility of the work that we publish. This form provides structure for consistency and transparency in reporting. For further information on Nature Research policies, see our Editorial Policies and the Editorial Policy Checklist.

## Statistics

For all statistical analyses, confirm that the following items are present in the figure legend, table legend, main text, or Methods section.

| n/a | Confirmed | |
|---|---|---|
| ☐ | ☒ | The exact sample size (*n*) for each experimental group/condition, given as a discrete number and unit of measurement |
| ☐ | ☒ | A statement on whether measurements were taken from distinct samples or whether the same sample was measured repeatedly |
| ☐ | ☒ | The statistical test(s) used AND whether they are one- or two-sided *Only common tests should be described solely by name; describe more complex techniques in the Methods section.* |
| ☒ | ☐ | A description of all covariates tested |
| ☒ | ☐ | A description of any assumptions or corrections, such as tests of normality and adjustment for multiple comparisons |
| ☐ | ☒ | A full description of the statistical parameters including central tendency (e.g. means) or other basic estimates (e.g. regression coefficient) AND variation (e.g. standard deviation) or associated estimates of uncertainty (e.g. confidence intervals) |
| ☐ | ☒ | For null hypothesis testing, the test statistic (e.g. *F*, *t*, *r*) with confidence intervals, effect sizes, degrees of freedom and *P* value noted *Give P values as exact values whenever suitable.* |
| ☒ | ☐ | For Bayesian analysis, information on the choice of priors and Markov chain Monte Carlo settings |
| ☒ | ☐ | For hierarchical and complex designs, identification of the appropriate level for tests and full reporting of outcomes |
| ☒ | ☐ | Estimates of effect sizes (e.g. Cohen's *d*, Pearson's *r*), indicating how they were calculated |

*Our web collection on statistics for biologists contains articles on many of the points above.*

## Software and code

Policy information about availability of computer code

| Data collection | No software was used for data collection. |
|---|---|
| Data analysis | All software packages used for analysis are cited in the Online Methods section and in the Supplementary Information and all used packages are publicly available: base caller: Bustard (Illumina); adapter trimming: leeHom (version used for this manuscript available at https://bioinf.eva.mpg.de/leehom/); mapping: Burrows-Wheeler Aligner (BWA, version 0.5.10-evan.9-1-g44db244); PCR duplicate removal: bam-rmdup (version 0.6.3, https://github.com/mpieva/biohazard-tools); handling BAM files: samtools (version 1.3.1), bedtools (version 2.24.0), bam-caller (https://github.com/bodkan/bam-caller, version 0.1); mtDNA contamination estimates: schmutzi (version 1.5.5); nuclear contamination estimates: ANGSD (version 0.929-27-ge7739a5), AuthentiCT (https://github.com/StephanePeyregne/AuthentiCT, version 1.0.0); f-statistics and qpGraph: ADMIXTOOLS (version 5.1) and R package admixr (version 0.7.1); genotype calls for Y-chromosome of F6-620: snpAD (version 0.3.4); haplogroup calling: yHaplo (version 1.0.18); PCA: smartpca from EIGENSOFT package; detecting archaic introgressed segments: admixfrog (version 0.5.6,  https://github.com/BenjaminPeter/admixfrog/). |

For manuscripts utilizing custom algorithms or software that are central to the research but not yet described in published literature, software must be made available to editors and reviewers. We strongly encourage code deposition in a community repository (e.g. GitHub). See the Nature Research guidelines for submitting code & software for further information.

## Data

Policy information about availability of data

All manuscripts must include a data availability statement. This statement should provide the following information, where applicable:

- Accession codes, unique identifiers, or web links for publicly available datasets
- A list of figures that have associated raw data
- A description of any restrictions on data availability

Newly produced sequence data of Bacho Kiro Cave specimens and Oase 1 are deposited in the European Nucleotide Archive (ENA, https://www.ebi.ac.uk/ena/browser/home) under the accession number PRJEB39134.

Comparative data of present-day human genomes from Simons Genome Diversity Project that were used in this study is available at: https://www.simonsfoundation.org/simons-genome-diversity-project/

Comparative data used in this study that includes genotypes of 2,109 ancient and 2,974 present-day individuals compiled from published studies is available in the EIGENSTRAT file format at: https://reich.hms.harvard.edu/downloadable-genotypes-present-day-and-ancient-dna-data-compiled-published-papers/ (version 37.2, released February 22, 2019).

To determine the Y chromosome haplogroups of male individuals in this study, we used the Y-haplogroup tree from the International Society of Genetic Genealogy (ISOGG, http://www.isogg.org, version: 13.38).

# Field-specific reporting

Please select the one below that is the best fit for your research. If you are not sure, read the appropriate sections before making your selection.

☒ Life sciences ☐ Behavioural & social sciences ☐ Ecological, evolutionary & environmental sciences

For a reference copy of the document with all sections, see nature.com/documents/nr-reporting-summary-flat.pdf

# Life sciences study design

All studies must disclose on these points even when the disclosure is negative.

| | |
|---|---|
| Sample size | The number of genomes analysed in this study was determined by identifying those specimens that had sufficient levels of ancient DNA preservation for downstream sequencing and analysis. Human specimens from this time period in Europe are extremely scarce, and in this study we screened seven specimens excavated recently from the Bacho Kiro Cave in Bulgaria: <br> - a human lower molar (F6-620) found in the upper part of the Layer J in the Main Sector of Bacho Kiro Cave <br> - four bone fragments (AA7-738, BB7-240, CC7-2289 and CC7-335) from Layer I in the Niche 1 sector <br> - a bone fragment from the Layer B in the Main Sector (F6-597) <br> - a bone fragment that was identified among the finds from excavations in the 1970s when it was retrieved in a position corresponding to the interface of Layers B/C (BK1653). <br> The bone fragments were initially identified as hominin based on the Zooarchaeology by Mass Spectrometry (ZooMS), and we screened them for ancient DNA preservation. Therefore, the sample size was predetermined based on the availability of the identified hominin specimens. <br> We also used the remaining bone powder of Oase 1 specimen from Romania from the 2015 study (Fu et al, Nature) and treated it with 0.5% hypochlorite solution in order to remove some of the present-day human and microbial contamination. <br> Given very low proportion of endogenous DNA and high levels of present-day human DNA contamination, we excluded the specimen F6-597 from downstream in solution hybridization captures and analyses. |
| Data exclusions | We used pre-established criteria in ancient DNA research of excluding sequences from the sequencing data that did not map to the human genome, sequences that were shorter than 35 base pairs and sequences mapping with a low mapping quality (< 25), all of which are excluded to avoid using sequences that are not endogenous to the individual sequenced. <br> Given the high contamination estimates and low nuclear DNA content, we excluded the libraries of the specimen F6-597 from nuclear captures and further downstream analyses. |
| Replication | The specimens sampled in this study were sampled on three different occasions. We produced in total 35 single-stranded DNA libraries from seven extracts of Bacho Kiro Cave specimens and 6 single-stranded libraries from three Oase 1 extracts, along with respective extraction and library negative controls, and distributed over six different experiments. The results of reproducibility of the data generation and analyses are reported across the tables in the Supplementary Information Section 2. <br> To allow the reproducibility of the downstream analyses, all filtering steps and the comparative data used are detailed in the Online Methods and the Supplementary Information of this study; also all of the sequence data obtained from Bacho Kiro Cave specimens and Oase 1 needed for replication of obtained results and conclusions of this study are deposited in the European Nucleotide Archive (ENA) under the accession number PRJEB39134. |
| Randomization | Randomization is not relevant to this study. We first determined DNA preservation in all ancient specimens selected for this study and then proceeded to analyse generated genome-wide data of all specimens that showed evidence of endogenous DNA preservation. |
| Blinding | Blinding was not relevant as we sampled ancient hominin specimens that were selected for this study based on their age and provenance, thus blinding would be inappropriate given the scarcity and value of the sampled material. Blinding in downstream data analyses was not relevant given that we analysed genome-wide data of eight specimens in relation to the publicly available datasets of present-day and ancient |

human genomes.

# Reporting for specific materials, systems and methods

We require information from authors about some types of materials, experimental systems and methods used in many studies. Here, indicate whether each material, system or method listed is relevant to your study. If you are not sure if a list item applies to your research, read the appropriate section before selecting a response.

## Materials & experimental systems

| n/a | Involved in the study |
|---|---|
| ✗ | ☐ Antibodies |
| ✗ | ☐ Eukaryotic cell lines |
| ☐ | ✗ Palaeontology and archaeology |
| ✗ | ☐ Animals and other organisms |
| ✗ | ☐ Human research participants |
| ✗ | ☐ Clinical data |
| ✗ | ☐ Dual use research of concern |

## Methods

| n/a | Involved in the study |
|---|---|
| ✗ | ☐ ChIP-seq |
| ✗ | ☐ Flow cytometry |
| ✗ | ☐ MRI-based neuroimaging |

## Palaeontology and Archaeology

| Specimen provenance | Specimen Oase 1 is deposited at the "Emil Racovita" Institute of Speleology in Bucharest, Romania. Permission was granted to Svante Pääbo of the MPI-EVA in September 2009 for the sampling of the Oase 1 for genetic analyses with the specimen being sampled on the September 28, 2009 in Bucharest, Romania. |
|---|---|
| | Excavation of the Bacho Kiro Cave was authorized by the Bulgarian Ministry of Culture, and delivered by NAIM-BAS: Nr124/11.05 2015; Nr225/28.04.2016; Nr47/02.05.2017; Nr99/17.04.2018/ Nr120/2019. Bacho Kiro Cave specimens were sampled in the clean room facility of the MPI-EVA in Leipzig, Germany in January and March 2018. |

| Specimen deposition | The Oase 1 specimen is deposited at "Emil Racovita" Institute of Speleology, Department of Geospeleology and Palaeontology, str. Frumoasa 31, Bucharest, Romania. |
|---|---|
| | The palaeontological material from the Bacho Kiro Cave is deposited at the National Museum of Natural History in Sofia, Bulgaria. |

| Dating methods | There are no new radiocarbon dates provided in this study. However, all the previously published radiocarbon dates were newly re-calibrated using the new calibration curve IntCal20 and are provided in the Supplementary Information Section 1, Table S1.1. |
|---|---|

✗ Tick this box to confirm that the raw and calibrated dates are available in the paper or in Supplementary Information.

| Ethics oversight | All approvals for specimen handling have been obtained from the relevant institutions: |
|---|---|
| | - for the Oase1 specimen, the permission was granted to Svante Pääbo of the MPI-EVA by the Emil Racovita Institute of Speleology, as national authority in caves study. |
| | - for the Bacho Kiro Cave specimens the permission was granted by the Bulgarian Ministry of Culture and the National Museum of Natural History (Sofia, Bulgaria) |

Note that full information on the approval of the study protocol must also be provided in the manuscript.

