## [Peer Review File · Nature]

Manuscript Title: Initial Upper Paleolithic humans in Europe had recent Neanderthal ancestry

Editorial Notes:

Reviewer Comments & Author Rebuttals**Reviewer Reports on the Initial Version:**Referee #1 (Remarks to the Author):

I found this is very interesting and extremely well written work, that follows a logical path of exploration of genomic data from several individuals excavated at the Bacho kiro IUP Paleolithic site in Bulgaria that can be some, if not the oldest modern human remains in the European continent. Besides the intrinsic interest of having new UP data that in many ways it is more scarce than Neanderthal data but also much more complex to interpret, the authors describe that these individuals (the oldest ones) bear signals of a quite recent Neanderthal hybridisation in their genealogical past. In addition, the Bacho Kiro individuals, although their ancestry is not present in modern Europeans (such as other, previous UP individuals sequenced), it shows ancestry links with early East Asian individuals (and consequently, to some extent, even to Amerindians), exemplifying the complex interactions that took place likely between 50 to 40,000 years ago among different groups across Eurasia. I don't have any criticism in the statistical analysis or the modelling

I think this is a work that deserves to be published and will like to be debated and cited. If anything, it is a pity the authors do not try to make a general inference of these complex early MH relationships, even if a bit speculative, in the line for instance, of Wong et al. (2017), *Genome Res* (Fig 8). I don't know if there are also potential links related for instance to lithic industry that could explain the broad pattern of ancestry found here (from East Asia to Eastern Europe); maybe this will deserve subsequent research, considering right now there are no more ancient remains with genomic data in between in this chronology.

Minor points: there are some typos in ref, 1,2, and 43 (journal issues and page numbers).

Referee #2 (Remarks to the Author):

The present manuscript by Hajdinjak et al. presents genomic data from some of the first anatomically modern human remains excavated in Europe from Bacho Kiro Cave in present-day Bulgaria. This is a great example how recent advances in palaeo- and archaeogenomics are making it possible to obtain DNA from such old specimen. Surprisingly, the Initial Upper Palaeolithic Europeans seem to show more genetic similarities to modern Asians than to modern Europeans. Other results are consistent with previous studies the IUP individuals had recent Neanderthal ancestors highlighting once more that Eurasian hominins mixed frequently which has been shown in various studies from the Denisova cave (coming from the same team) and we still see signatures of a subset of these admixture events in modern non-Africans. Furthermore, the study analyzes the locations of archaic segments across the genome which confirms the presence of multiple independent admixture events and that the IUP individuals show similar genomic regions without archaic ancestry suggesting that purifying selection removed Neanderthal ancestry from these regions within very few generations.

The article is well written and the conclusions are based on established methodology or on methods that are presented elsewhere. The results regarding archaic admixture mostly confirm previous results. The most significant finding is the genetic similarity between the IUP Europeans and modern eastern Eurasian groups. Earlier studies on contemporary (or slightly later) individuals like Ust-Ishim and Oase1 have revealed that those individuals do not share an excess of affinity to either modern

eastern or western Eurasians. It is interesting that Bacho Kiro appears to represent a potentially earlier European stratum with different ancestry. One problem here is the limitation of available data to compare these individuals to. Were these groups with Asian affinity later replaced by the symmetrically related groups or did they live side by side without mixing substantially? (This seems unlikely considering the results about frequent mixtures between all different hominins.) Could the IUP Bacho Kiro individuals even represent an earlier Out-of-Africa event which spread across Eurasia, Oceania and the Americas but was replaced by later migrations in western Eurasia? The authors (understandably) are careful when interpreting the limited data available so far. This limits the significance of the present study but it makes the conclusions solid and leaves the questions for later research.

Minor comments:

Most of the manuscript modestly calls the generated data “genome-wide” to indicate the SNP capture procedure used to produce it, whereas the title uses “genomes”.

L125/126: I guess the remains could also belong to identical twins. Less likely than them being the same individual but a possible explanation.

It would be useful to see the values behind figure 2A in a supplementary table.

Numbering for Extended Data Figures in the main document seems to be out of order.

Please check the resolution of figures in the supplementary material, e.g. SI 5.

Referee #3 (Remarks to the Author):

This is a nice, interesting and detailed paper that based on the molecular analyses of the recently published Bacho Kiro fossils from Bulgaria provides additional genetic data about modern humans-Neanderthals interaction and offers some possible interpretations of the evolutionary scenario where these interactions took place. As such, I find the paper of interest for the scientific community as it adds to the growing body of genetic evidence that will eventually help us to build a more precise picture of such a complex topic like the biological and cultural interaction of *Homo sapiens* and *Homo neanderthalensis* (demography, dispersals, competition, biogeography, hybridization...). However, it is not clear to me to what extent this paper is providing significant new, novel or ground-breaking information to be published in a journal like Nature instead of in a more specialized publication.

Overall, the paper comes to support the previous notion about Neanderthals mixing with *H. sapiens*, mostly in Eastern Europe where populations like Bacho Kiro and the already known Oase 1 hold higher proportions of introgressed Neanderthal DNA than other archaeological and present-day *Homo sapiens* groups, and from a relatively close ancestor (“less than about six generations back in their family tree”). I confess my worry about what it seems to be a common trend in recent times, the overinterpretation of the genetic data to push the generation of a headline that offers general conclusions about immensely complex topics (demography, dispersals and interactions) of populations based on literally the analysis of single specimens and only from one line of evidence (genetics). The “novelty” in this paper is stretched from the “stark contrast” (sic) of the 45kyr Ust’Ishim and 40 kyr Oase 1 specimens “that did not contribute **substantially**” (note the substantially nuance- it does not say none although later in the paper is removed) to later populations”, whereas the IUP Bacho Kiro “individuals are more related to present-day and ancient populations with East Asian ancestry than to later West Eurasian ones”. Is the sample size enough and the statistical comparison robust enough to make such a general statement? I wonder how based on the analysis of three isolated samples we can make inferences about ancestry of archaic and present day populations that span more than 40,000 years of evolution, genetic drift, migration and selection that we cannot obviously apprehend with the data presented here. As an example, the

relationship of the Oase1 and Ust'Ishim individual to subsequent Eurasian populations is majorly based on the analysis of the 40,000 year old Tianyuan individual from the giant China (page 3, line 67-72), oversimplifying such a complex topic beyond the mere curiosity of Bacho Kiro having overall less affinities than the other two specimens with present day groups. Simply presenting "more alleles" shared with present day groups than the other two populations, without going down to discriminate the type of alleles and the type of selection those alleles can be subject to in such a wide time range (tens and hundreds of thousands of years) is leading us to an oversimplification that is misleading for the general public. It looks like if the analysis of a single specimen can solve the complex study of hominins demography and dispersals without even having to integrate the profuse archaeological and paleobiological evidence. In their final paragraphs authors conclude the existence of "several differentially related modern human populations" in Eurasia that show "no affinities to later populations" (we now eliminate the "substantial" nuance) just based on the genetic analysis of a few specimens from the same locality and without any archaeological or fossil support beyond a mere mention of the Kuhn and Zwyns paper in support of a possible northern dispersal.

I believe this is an excellent genetic contribution that should be published in a good SCI journal but escaping from the thirst of sensationalism, which is not needed. We are dangerously crossing the line between scientific literature to simply literature.

Author Rebuttals to Initial Comments:

Referees' comments:

Referee #1 (Remarks to the Author):

I found this is very interesting and extremely well written work, that follows a logical path of exploration of genomic data from several individuals excavated at the Bacho kiro IUP Paleolithic site in Bulgaria that can be some, if not the oldest modern human remains in the European continent. Besides the intrinsic interest of having new UP data that in many ways it is more scarce than Neandertal data but also much more complex to interpret, the authors describe that these individuals (the oldest ones) bear signals of a quite recent Neandertal hibridisation in their genealogical past. In addition, the Bacho Kiro individuals, although their ancestry is not present in modern Europeans (such as other, previous UP individuals sequenced), it shows ancestry links with early East Asian individuals (and consequently, to some extent, even to Amerindians), exemplifying the complex interactions that took place likely between 50 to 40,000 years ago among different groups across Eurasia. I

don't have any criticism in the statistical analysis or the modelling

I think this is a work that deserves to be published and will like to debated and cited. If anything, it is a pity the authors do not try to make a general inference of these complex early MH relationships, even if a bit speculative, in the line for instance, of Wong et al. (2017), Genome Res (Fig 8). I don't know if there are also potential links related for instance to lithic industry that could explain the broad pattern of ancestry found here (from East Asia to Eastern Europe); maybe

this will deserve subsequent research, considering right now there are no more ancient remains with genomic data in between in this chronology.

As suggested by the reviewer, we have now provided more context to the genomic findings by adding archaeological sites where IUP lithic assemblages have been found to the Figure 1 as well as a Supplementary section (Supplementary Information 1) discussing the archaeology. We highlight the broad distribution of IUP assemblages from central Europe and the Levant to Mongolia and northwest China and how this may be connected to the genetic patterns showing that the IUP Bacho Kiro individuals contributed ancestry to later populations in East Asia. We also discuss this broader context of the genetic findings in the main text (lines 78 to 88 and 240 to 254).

Minor points: there are some typos in ref, 1,2, and 43 (journal issues and page numbers).

The typos are now corrected.

Referee #2 (Remarks to the Author):

The present manuscript by Hajdinjak et al. presents genomic data from some of the first anatomically modern human remains excavated in Europe from Bacho Kiro Cave in present-day Bulgaria. This is a great example how recent advances in palaeo- and archaeogenomics are making it possible to obtain DNA from such old specimen. Surprisingly, the Initial Upper Palaeolithic Europeans seem to show more genetic similarities to modern Asians than to modern Europeans. Other results are consistent with previous studies the IUP individuals had recent Neanderthal ancestors highlighting once more that Eurasian hominins mixed frequently which has been shown in various studies from the Denisova cave (coming from the same team) and we still see signatures of a subset of these admixture events in modern non-Africans.

Furthermore, the study analyzes the locations of archaic segments across the genome which confirms the presence of multiple independent admixture events and that the IUP individuals show similar genomic regions without archaic ancestry suggesting that purifying selection removed Neanderthal ancestry from these regions within very few generations.

The article is well written and the conclusions are based on established methodology or on methods that are presented elsewhere. The results regarding archaic admixture mostly confirm previous results.

We have partially re-written the main text of our manuscript to highlight the novel insights provided by the IUP Bacho Kiro Cave individuals. Firstly, there is only one previous example of an early modern human with close Neandertal ancestors (*Oase1*; Fu et al, Nature, 2015). This single individual could have been an exceptional case. It also lacks an archaeological context and is only inferred to be associated with the IUP due to its age. Here we show that multiple individuals who are clearly associated with the IUP all had recent Neandertal relatives. Thus, we show that frequent mixing between the earliest modern humans outside of Africa and Neandertals is likely to have been the rule rather than the exception.

The most significant finding is the genetic similarity between the IUP Europeans and modern eastern Eurasian groups. Earlier studies on contemporary (or slightly later) individuals like Ust-Ishim and Oase1 have revealed that those individuals do not share an excess of affinity to either modern eastern or western Eurasians. It is interesting that Bacho Kiro appears to represent a potentially earlier European stratum with different ancestry. One problem here is the limitation of available data to compare these individuals to. Were these groups with Asian affinity later replaced by the symmetrically related groups or did they live side by side without mixing substantially? (This seems unlikely considering the results about frequent mixtures between all different hominins.) Could the IUP Bacho Kiro individuals even represent an earlier Out-of-Africa event which spread across Eurasia, Oceania and the Americas but was replaced by later migrations in western Eurasia? The authors (understandably) are careful when interpreting the limited data available so far. This limits the significance of the present study but it makes the conclusions solid and leaves the questions for later research.

As the reviewer indicates, we prefer to refrain from speculation and restrict ourselves to the inferences that the genomic data allow (but see comments from Referee #3). However, to put our results into an archaeological context, we now show the geographical distribution of IUP lithic assemblages in Figure 1. They extend to Mongolia and thus match the population genetic relationships we observe between the IUP Bacho Kiro individuals and later populations in East Asia as well as the 35,000-year-old *GoyetQ116-1* in Belgium (Fig. 2C and D, Tables S5.12-S5.14, Fig. S6.4-S6.6). The latter observation demonstrates that the IUP Bacho Kiro individuals contributed to some later populations in Europe too. However, this contribution had been replaced (at least to an extent to not be detectable with current methods) by subsequent migrations into West Eurasia by at least 38,000 years cal. BP (*Kostenki14*) and 35,000 years cal. BP in Bacho Kiro Cave itself (*BK1653*).

The reviewer raises the interesting possibility that the *Oase1* and *Ust'-Ishim* individuals may represent later populations distinct from the Bacho Kiro IUP population as there is no indication that they contributed to later populations. This is to our best understanding not possible to resolve at the

moment. It must await future discoveries of well-dated early modern humans in western Eurasia.

Minor comments:

Most of the manuscript modestly calls the generated data “genome-wide” to indicate the SNP capture procedure used to produce it, whereas the title uses “genomes”.

We prefer the shorter form in the title. We consider that we have generated representative, genome-wide data (not sequenced entire genomes) but it is still the “genomes” that reveal the insights we describe.

L125/126: I guess the remains could also belong to identical twins. Less likely than them being the same individual but a possible explanation.

We now acknowledge the unlikely possibility that the specimens F6-620 and AA7-738 could stem from identical twins on lines 135 to 136.

It would be useful to see the values behind figure 2A in a supplementary table.

We have added additional tables to the Supplementary Information 5 (previously Supplementary Information 4). They provide the values of f_3 -statistics for all IUP Bacho Kiro individuals considered together (Fig. 2A, Table S5.1, page 46), each of the three IUP Bacho Kiro Cave individuals separately (Fig. S5.1, Tables S5.2.-S5.4, pages 47-49), and BK1653 individual (Extended Data Fig. 3B, Table S5.5, page 50).

Numbering for Extended Data Figures in the main document seems to be out of order.

We have corrected the numbering for the Extended Data Figures in the main text.

Please check the resolution of figures in the supplementary material, e.g. SI 5.

We have improved the resolution of the figures in Supplementary Information 6 (previously Supplementary Information 5), Figures S6.1-S6.6, pages 76-82.

Referee #3 (Remarks to the Author):

This is a nice, interesting and detailed paper that based on the molecular analyses of the recently published Bacho Kiro fossils from Bulgaria provides additional genetic data about modern humans-Neanderthals interaction and offers some possible interpretations of the evolutionary scenario where these interactions took place. As such, I find the paper of interest for the scientific community as it adds to the growing body of genetic evidence that will eventually help us to build

a more precise picture of such a complex topic like the biological and cultural interaction of *Homo sapiens* and *Homo neanderthalensis* (demography, dispersals, competition, biogeography, hybridization...).

We thank the reviewer for these positive comments and the constructive comments below.

However, it is not clear to me to what extent this paper is providing significant new, novel or ground-breaking information to be published in a journal like Nature instead of in a more specialized publication.

Our manuscript presents two major new advances in our understanding of early modern humans in Eurasia:

1. The three early Bacho Kiro Cave individuals represent an early expansion of modern humans into Europe that was previously unknown in the genetic record, with implications for the broader out-of-Africa expansion. They also provide the first genome-wide data from individuals directly associated with Initial Upper Palaeolithic assemblages. The surprising finding that the early Bacho Kiro individuals do not show genetic links to later Europeans but instead to East Eurasians, together with our detailed modelling, reveals that archaeological hypotheses about the importance of the Initial Upper Palaeolithic were correct, and may have shared a common history stemming from an early into-Eurasia expansion from Africa or Southwest Asia. This is a new insight that rewrites previous genetic models for the modern human expansion out-of-Africa and into Eurasia.

2. Interbreeding between Neandertals and the first modern humans in Europe was common. This is a longstanding question in genetics, archaeology, and palaeoanthropology that was believed by many to have been falsified by the absence of excess Neandertal ancestry in present-day Europeans relative to Asians (Green et al. 2010), and in the ancient DNA that was previously available. A previous single ~40-ky-old individual from Europe (*Oase 1*, Fu et al. 2015) was shown to have a recent Neandertal ancestor, but this could not be excluded to have been a chance find. In our paper, we demonstrate that three of three early individuals from Bacho Kiro Cave had very recent Neandertal ancestors, demonstrating that interbreeding, while not necessarily ubiquitous, must have been common in the first modern humans in Europe, with four of four individuals known to have overlapped with Neandertals in Europe now showing recent interbreeding.

Overall, the paper comes to support the previous notion about Neanderthals mixing with *H. sapiens*, mostly in Eastern Europe where populations like Bacho Kiro and the already known *Oase 1* hold higher proportions of introgressed Neandertal DNA than other archaeological and

present-day Homo sapiens groups, and from a relatively close ancestor (“less than about six generations back in their family tree”).

We agree with this, but note that while the previous notion may have been held by many in palaeoanthropology and archaeology, it was not accepted among geneticists. This is because no larger proportion of Neandertal DNA has been observed in any individuals in Europe apart from the single *Oase 1* individual. The Bacho Kiro evidence now resolves these models across the fields, revealing that while admixture was common in the early modern humans of Europe, they contributed only limited ancestors to later human groups, and thus the "European" Neandertal ancestry was not passed on in a detectable way.

I confess my worry about what it seems to be a common trend in recent times, the overinterpretation of the genetic data to push the generation of a headline that offers general conclusions about immensely complex topics (demography, dispersals and interactions) of populations based on literally the analysis of single specimens and only from one line of evidence (genetics).

We restrict our inferences to the areas where genomic information can provide insights, *i.e.* population relationships and individual relationship.

The “novelty” in this paper is stretched from the “stark contrast” (sic) of the 45ky Ust’Ishim and 40 kyr Oase 1 specimens “that did not contribute substantially (note the substantially nuance- it does not say none although later in the paper is removed) to later populations”, whereas the IUP Bacho Kiro “individuals are more related to present-day and ancient populations with East Asian ancestry than to later West Eurasian ones”.

We agree with the reviewer’s notion that such statements of contributions of ancestry to later populations are limited, and do not take complicated demographic models into account. We can show clearly that the early Bacho Kiro group cannot have contributed substantially to immediately later

populations 40-20 ky BP in Europe, as for example the excess Neandertal ancestry is not observed, and additional analyses confirm this. However, these individuals are still genetically significantly closer to later populations in East Asia than in West Eurasia. With the current data, we cannot confidently say whether or not *Oase1* and *Ust'Ishim* might have contributed to East Asian or any other present-day populations further removed by time. However, to the limits of our resolution, we do not find these individuals being detectably closer to any ancient or present-day population. In the revised version of the manuscript, we have de-emphasized statements about not contributing or contributing to later populations.

We say “substantially” to signal that every statement are limited by the level of detection possible. We realize that “substantially” may be unclear and have now replaced it with “detectably”.

Is the sample size enough and the statistical comparison robust enough to make such a general statement? I wonder how based on the analysis of three isolated samples we can make inferences about ancestry of archaic and present day populations that span more than 40,000 years of evolution, genetic drift, migration and selection that we cannot obviously apprehend with the data presented here.

In the paper, we leverage the full ancient genetic record that also includes over 40 individuals from Upper Palaeolithic Europe. However, we agree with the reviewer that the interpretation is most clear when not extrapolated across large temporal spans, and in the revised version we have de-emphasized the comparison to present-day populations.

As an example, the relationship of the Oase1 and Ust'Ishim individual to subsequent Eurasian populations is majorly based on the analysis of the 40,000 year old Tianyuan individual from the giant China (page 3, line 67-72), oversimplifying such a complex topic beyond the mere curiosity of Bacho Kiro having overall less affinities than the other two specimens with present day groups. Simply presenting “more alleles” shared with present day groups than the other two populations, without going down to discriminate the type of alleles and the type of selection those alleles can be subject to in such a wide time range (tens and hundreds of thousands of years) is leading us to an oversimplification that is misleading for the general public.

Here the reviewer lacks an understanding of genetic analyses of ancestry. We incorporate a careful statistical approach, incorporating a Weighted Block Jackknife approach across 5 centimorgan segments of the genome (Busing *et al*, Statistics and Computing, 1999, <https://link.springer.com/article/10.1023/A:1008800423698>) to confirm that the sharing of more alleles (or not) between different ancient individuals have convincing statistical support.

The relationship of *Ust'-Ishim* and *Oase 1* to the subsequent Eurasian populations is not based on only the comparison to the Tianyuan individual from China. These relationships are inferred by comparing *Ust'-Ishim* and *Oase 1* to all available subsequent Eurasian ancient and present-day populations. The statistical approaches used are mentioned above and described in the paper.

The IUP Bacho Kiro individuals do not have less, but more, affinity than *Ust'-Ishim* and *Oase 1* to later populations. In our opinion, this is not a “mere curiosity”. It is the first demonstration that IUP population contributed genetically to later populations.

Nevertheless, we agree that it is most useful to compare specimens close in time, and so have de-emphasized the comparison between ancient and present-day populations.

It looks like if the analysis of a single specimen can solve the complex study of hominins demography and dispersals without even having to integrate the profuse archaeological and paleobiological evidence.

Single nuclear genomes provide information not only about the individuals, but also about tens of thousands of their ancestors. A single genome is not to compare to, e.g., the morphology of a single tooth find, or the shape of a single hand axe, but perhaps more aptly to a large book written in an extinct language to a linguist. It also differs from mtDNA and Y-chromosomes, which are inherited as single units from one parent to the next generation.

However, in the revised version we have included additional information and integration with the archaeological and paleoanthropological record. We make no claims to solve hominin demography and dispersal beyond the specific, but central, questions discussed in the text about the first modern humans in Europe and whether they interbred with Neandertals.

In their final paragraphs authors conclude the existence of “several differentially related modern human populations” in Eurasia that show “no affinities to later populations” (we now eliminate the “substantial” nuance) just based on the genetic analysis of a few specimens from the same locality and without any archaeological or fossil support beyond a mere mention of the Kuhn and Zwyns paper in support of a possible northern dispersal.

I believe this is an excellent genetic contribution that should be published in a good SCI journal but escaping from the thirst of sensationalism, which is not needed. We are dangerously crossing the line between scientific literature to simply literature.

The archaeology, palaeobiology, lithics, ornaments and radiocarbon dating of Bacho Kiro Cave, as well as how these finds fit with the IUP assemblages found at other archaeological sites, have been described in detail in the recent publications of Hublin *et al.*, Nature, 2020 (<https://www.nature.com/articles/s41586-020-2259-z>) and Fewlass *et al.*, Nature Ecology and Evolution, 2020 (<https://www.nature.com/articles/s41559-020-1136-3>). This work is complementary to those two papers. Nevertheless, we have now expanded on these things by adding an additional Supplementary Information section on the archaeology (novel Supplementary Information 1).

As outlined in our first response to the reviewer, this manuscript answers two central questions about the first modern humans in Europe: who they were and whether they interbred with Neandertals.

Reviewer Reports on the First Revision:

Referee #1 (Remarks to the Author):

I was in general positive in the previous version of this manuscript, as it is easy to recognize the importance of having genome-wide data from Bacho-Kiro, one of the most important sites of early modern humans in Europe. Nevertheless, the authors have even improved the scope and clarity of their findings, provided, as I suggested, a broader range of interpretation and a larger archaeological context. I think there are two important points in this work that likely will need to be followed up in the future and that seems relevant to different fields studying the human past; the fact that there was a common ancestry in the IUP from Eastern Europe to East Asia -which is, of course, an enormous and diverse geographical area- and the evidence that different early modern humans into Europe, with different ancestries, have a Neandertal ancestor few generations ago. This supports an intriguing idea that maybe Neandertals -or part of them- were in fact not replaced but literally "absorbed" by the MH incomers.

Minor typos:

-Line 50 (Abstract): that that

-Refs 15,19,21,41 and 42 all refer to the same journal that is written in two different forms (both of them, I am afraid, incorrect)

-Ref. 34, should be Nature 570: 182-188, not Nature 1.

Referee #2 (Remarks to the Author):

I thank the authors for preparing this revised version of the manuscript. It represents an improvement over an already well-written manuscript. I still think this is an interesting study which has its value and will become an important part of the literature similar to previous genomic studies on early Eurasian modern humans (e.g. Fu et al PNAS 2013; Fu et al Nature 2014, 2015, 2016; Seguin-Orlando et al Science 2014; Yang et al Curr Biol 2017). There seem to be different opinions on the novelty of the findings in this manuscript. Since I am working in this field, I would read and cite this article independent of the journal it is published in as I consider it valuable, interesting and important.

In addition to the generation of genomic data from the first modern humans in Europe in general, I still think the main novel finding is the line of ancestry that the IUP Bacho Kiro individuals seem to represent. They show stronger similarities to later Eastern Eurasians than Western Eurasians which is an observation that will motivate further research. At present, we only have this intriguing

observation from f statistics and an admixture graph with high levels of private drift for several individuals indicating that they only represent distant relatives of the mixing groups. It will be interesting to see if remains excavated and sequenced in the future will help us understand this pattern in a better way. The combination with lithic assemblages does indicate potential connections.

The second main finding appears to be the recent Neandertal ancestor. Only one AMH with a recent Neandertal ancestor has been found so far. The abstracts reads:

"Moreover, we find that all three IUP Bacho Kiro individuals had Neandertal ancestors a few generations back in their family history, suggesting that the first modern humans that arrived in Europe mixed frequently with Neandertals."

We can compare this to a previous study by the same team (Slon et al Nature 2018) where they did not sequence an anatomically modern human but they still reach similar conclusions. The abstract of that article reads:

"The finding of a first-generation Neanderthal–Denisovan offspring among the small number of archaic specimens sequenced to date suggests that mixing between Late Pleistocene hominin groups was common when they met."

And the conclusions from the same article:

"It is notable that one direct offspring of a Neanderthal and a Denisovan (Denisova 11) and one modern human with a close Neanderthal relative (Oase 1) have been identified among the few individuals from whom DNA has been retrieved and who lived at the time of overlap of these groups (Fig. 1). In conjunction with the presence of Neanderthal and Denisovan DNA in ancient and present-day people^{2,5,8,13,16,17,25,26,27}, this suggests that mixing among archaic and modern hominin groups may have been frequent when they met."

Finally, I do not agree with the notion that "genome-wide" and "genome" can be used in the title interchangeably. If the authors are concerned with brevity, they could have also used "DNA" which is even shorter and also would imply something similar. I agree that one can argue what a "genome" is when one is doing re-sequencing of short fragments which will never have the resolution to reconstruct a full genome. But I also think that the title should be accurate about the type of data (as used elsewhere across the ms) and that this does not reduce the quality of the data or the article.

Referee #4 (Remarks to the Author):

This article presents the paleogenomic analysis of several of the newly excavated modern human bone fragments from Bacho Kiro, recently published as the earliest evidence of Upper Paleolithic modern humans in Europe. Results show that the earliest of these individuals, associated with the Initial Upper Paleolithic, are more closely related to modern-day East Eurasians than to Western Eurasians, and they also show evidence of recent admixture with Neanderthals. Results from a later individual from the same site shows instead a possible contribution to later Western Eurasian populations.

This work is original in that well established methods developed by the authors for the analysis of ancient DNA are being applied to newly excavated / discovered specimens.

The data are presented clearly and extensively and the authors are among the most experienced experts in this field.

The significance of this work appears to me to be of an incremental nature rather than a major breakthrough. The taxonomic identification of these specimens as modern humans was already established previously through both morphological and mt DNA analysis (Hublin et al. 2020).

Evidence of interbreeding between ancient human taxa / lineages has been extensively reported and such admixture is now considered to have been much more common than previously thought, and perhaps the norm in situations where contact occurred. Certainly the results presented point to interesting potential relationships between Late Pleistocene modern human groups and present day populations in Eurasia. Rather than taking these at face value, however, and given the scarcity of genetic data as well as the potential problems of contamination, such proposed relationships should form the basis of hypotheses to be tested through in-depth integration with the archaeological and fossil record, as well as through future additional genetic and paleogenetic studies.

Instead the interpretations presented here consider other aspects of the record only superficially, and the conclusions seem rather simplistic.

Additional points:

The authors refer to the Bacho Kiro remains as the earliest modern humans in the European continent. This should be amended to 'earliest Upper Paleolithic' or 'earliest Late Pleistocene' modern humans. The discoveries from Apidima cave published recently in this journal suggest that a much earlier modern human dispersal, previously documented from the Near East, also reached Europe.

Figure 2 B and C are unclear to non-specialists. What exactly is being plotted here? The legend of the figure should be clarified.

Referee #5 (Remarks to the Author):

In my opinion reviewer #3's concerns have been sufficiently addressed.

- Rev. 3 worried of the ground-breaking nature of the data presented in the manuscript to warrant a publication in Nature. The authors (now) provide clear evidence for the ground-breaking nature of their data warranting publication in Nature.

o They provide the first genome wide data for the three oldest (circa 45 ky BP) Homo sapiens found in Europe. These three oldest Homo sapiens are also the first ones of that time-period time found in direct association to an archaeological context that can give indications on the cultural identity of these individuals. Before this study, only two European Homo sapiens individuals dated to circa 40,000 years (Oase 1 and Kostenki 14) were (genome-wide) genetically studied and none of them were found in an archaeological context. From their analysis of genome-wide new data, they show that these early modern Europeans "are more closely related to present-day and ancient populations in East Asia and the Americas than to later west Eurasians". They now observe in the conclusion that the genetic make-up of these earliest European Homo Sapiens from Bulgaria "is consistent with the fact that IUP archaeological assemblages are found from central and eastern Europe to present-day Mongolia, and a putative IUP dispersal that reached from eastern Europe to East Asia". They refrain from mentioning such an observation in the abstract, and this is legitimate given the work that need to be done to consolidate the knowledge of the IUP from a cultural and genetic point of view.

o They found that each of the three oldest individuals "had a Neandertal ancestor a few generations back in their family history". Before only one individual with mixed biological background was known and one could not exclude the possibility that it was a chance find. The analysis of three new individuals out of a total of 5, and 4 of these showing mixed ancestry, is indeed pointing toward an interbreeding pattern with the oldest homo sapiens population in Europe. The abstract terminates with suggesting that "the first modern humans that arrived in Europe mixed frequently with Neandertals".

- Rev. 3 also worries about "the overinterpretation of genetic data" and the offering "general conclusions about immensely complex topics".

o In my opinion, the fact that the authors refrain to refer in the abstract (or even in the title) to the observation made in the conclusion [about the consistency of the genetic signal with “the fact that IUP archaeological assemblages are found from central and eastern Europe to present-day Mongolia, and a putative IUP dispersal that reached from eastern Europe to East Asia”] is showing that they are careful to not overinterpret the data.

o They study a small number of individuals because it is still the start of such analysis and no other skeletal remains from that time period are available (hence also the ground-breaking nature of the data). Also, the genome of each of these individual is an archive of the genome of (“ten of thousands” say the authors in their rebuttal) of their ancestors. I agree with the authors that “they make no claims to solve hominin demography and dispersal beyond the specific, but central, question discussed in the text about the first modern humans in Europe and whether they interbred with Neandertals” (see authors rebuttal).

About “frequent” interbreeding, it remains unclear to me if the analyzed sampled could be biased against non-hybrids. Could it be that the individuals that entered the archaeological record were preferably of mixed ancestry... because they were less fit than others/because their cadaver were treated/taken care of differently?

It also remains unclear to me where exactly interbreeding happen because only the location of discovery of the analyzed human remains is known. The exact location of the interbreeding event is unknown and could maybe have happen in a different area, maybe even outside of current-days Romania/Bulgaria. Could it have happened outside Europe?

Overall, in my opinion the paper warrants publication in Nature. It is a paper that will be of immediate interest to the paleogeneticists, archeogeneticist but also archaeologists and paleoanthropologists. It will likely be abundantly cited.

Author Rebuttals to First Revision:

We would like to thank all reviewers for their positive comments and their highly constructive feedback.

Referees' comments:

Referee #1 (Remarks to the Author):

I was in general positive in the previous version of this manuscript, as it is easy to recognize the importance of having genome-wide data from Bacho-Kiro, one of the most important sites of early modern humans in Europe. Nevertheless, the authors have even improved the scope and clarity of their findings, provided, as I suggested, a broader range of interpretation and a larger archaeological context. I think there are two important points in this work that likely will need to be followed up in the future and that seems relevant to different fields studying the human past; the fact that there was a common ancestry in the IUP from Eastern Europe to East Asia - which is, of course, an enormous and diverse geographical area- and the evidence that different early modern humans into Europe, with different ancestries, have a Neandertal ancestor few generations ago. This supports an intriguing idea that maybe Neandertals -or part of them- were in fact not replaced but literally "absorbed" by the MH incomers.

Minor typos:

-Line 50 (Abstract): that that

Now fixed.

-Refs 15,19,21,41 and 42 all refer to the same journal that is written in two different forms (both of them, I am afraid, incorrect)

Now corrected to journal abbreviation *Proc. Natl. Acad. Sci. U.S.A* in all mentioned references.

-Ref. 34, should be Nature 570: 182-188, not Nature 1.

The reference 34 now corrected to: Sikora, M. *et al.* The population history of northeastern Siberia since the Pleistocene. *Nature* 570, 182-188, doi: 10.1038/s41586-019-1279-z (2019).

Referee #2 (Remarks to the Author):

I thank the authors for preparing this revised version of the manuscript. It represents an improvement over an already well-written manuscript. I still think this is an interesting study which has its value and will become an important part of the literature similar to previous genomic studies on early Eurasian modern humans (e.g. Fu et al PNAS 2013; Fu et al Nature 2014, 2015, 2016; Seguin-Orlando et al Science 2014; Yang et al Curr Biol 2017). There seem to be different opinions on the novelty of the findings in this manuscript. Since I am working in this field, I would read and cite this article independent of the journal it is published in as I consider it valuable, interesting and important.

In addition to the generation of genomic data from the first modern humans in Europe in general, I still think the main novel finding is the line of ancestry that the IUP Bacho Kiro individuals seem to represent. They show stronger similarities to later Eastern Eurasians than Western Eurasians which is an observation that will motivate further research. At present, we only have this intriguing observation from f statistics and an admixture graph with high levels of private drift for several individuals indicating that they only represent distant relatives of the mixing groups. It will be interesting to see if remains excavated and sequenced in the future will help us understand this pattern in a better way. The combination with lithic assemblages does indicate potential connections.

The second main finding appears to be the recent Neandertal ancestor. Only one AMH with a recent Neandertal ancestor has been found so far. The abstracts reads:

“Moreover, we find that all three IUP Bacho Kiro individuals had Neandertal ancestors a few generations back in their family history, suggesting that the first modern humans that arrived in Europe mixed frequently with Neandertals.”

We can compare this to a previous study by the same team (Slon et al Nature 2018) where they did not sequence an anatomically modern human but they still reach similar conclusions. The abstract of that article reads:

“The finding of a first-generation Neanderthal–Denisovan offspring among the small number of archaic specimens sequenced to date suggests that mixing between Late Pleistocene hominin groups was common when they met.”

And the conclusions from the same article:

“It is notable that one direct offspring of a Neanderthal and a Denisovan (Denisova 11) and one modern human with a close Neanderthal relative (Oase 1) have been identified among the few individuals from whom DNA has been retrieved and who lived at the time of overlap of these groups (Fig. 1). In conjunction with the presence of Neanderthal and Denisovan DNA in ancient and present-day people^{2,5,8,13,16,17,25,26,27}, this suggests that mixing among archaic and modern hominin groups may have been frequent when they met.”

Finally, I do not agree with the notion that “genome-wide” and “genome” can be used in the title interchangeably. If the authors are concerned with brevity, they could have also used “DNA” which is even shorter and also would imply something similar. I agree that one can argue what a “genome” is when one is doing re-sequencing of short fragments which will never have the resolution to reconstruct a full genome. But I also think that the title should be accurate about the type of data (as used elsewhere across the ms) and that this does not reduce the quality of the data or the article.

As suggested by the reviewer and the editor, we have now provided an alternative shorter title of the manuscript that does not include the word “genomes”.

Referee #4 (Remarks to the Author):

This article presents the paleogenomic analysis of several of the newly excavated modern human bone fragments from Bacho Kiro, recently published as the earliest evidence of Upper Paleolithic modern humans in Europe. Results show that the earliest of these individuals, associated with the Initial Upper Paleolithic, are more closely related to modern-day East

Eurasians than to Western Eurasians, and they also show evidence of recent admixture with Neanderthals. Results from a later individual from the same site shows instead a possible contribution to later Western Eurasian populations.

This work is original in that well established methods developed by the authors for the analysis of ancient DNA are being applied to newly excavated / discovered specimens.

The data are presented clearly and extensively and the authors are among the most experienced experts in this field.

The significance of this work appears to me to be of an incremental nature rather than a major breakthrough. The taxonomic identification of these specimens as modern humans was already established previously through both morphological and mtDNA analysis (Hublin et al. 2020).

Even though the morphological analyses of the human molar *F6-620* presented in *Hublin et al., 2020* clearly assigned the tooth to modern humans (with an exception of the middle trigonid crest), the remaining hominin fragments are morphologically undiagnostic. While we agree that the morphological and mtDNA analyses attributed the specimens to modern humans in *Hublin et al., 2020*, we caution that the mtDNA is a single locus inherited from one parent to the next generation and does not fully resolve the identification of these specimens. The best example is the recent discovery by Slon et al, 2018, where *Denisova 11* individual was first attributed to Neandertals based on the mtDNA analyses (see Brown et al, 2016), but turned out to be a first-generation offspring of Neandertals and Denisovans. Given that IUP Bacho Kiro Cave individuals overlap in time with some of the last Neandertals in Europe, it was still plausible that they might have larger contribution of Neandertal DNA to their nuclear genomes – which we indeed found to be the case.

Furthermore, nuclear DNA from these individuals does not only inform us of their relationship with Neandertals, but allowed us to identify them as belonging to a population of modern humans in Palaeolithic Europe that was previously unknown from the genetic record – something which was not possible from only the mtDNA genomes.

Evidence of interbreeding between ancient human taxa / lineages has been extensively reported and such admixture is now considered to have been much more common than previously thought, and perhaps the norm in situations where contact occurred. Certainly the results presented point to interesting potential relationships between Late Pleistocene modern human groups and present day populations in Eurasia. Rather than taking these at face value, however, and given the scarcity of genetic data as well as the potential problems of contamination, such proposed relationships should form the basis of hypotheses to be tested through in-depth

integration with the archaeological and fossil record, as well as through future additional genetic and paleogenetic studies.

Instead the interpretations presented here consider other aspects of the record only superficially, and the conclusions seem rather simplistic.

We agree with the reviewer that there is increasing evidence of interbreeding between and among ancient and archaic human groups. However, so far, only one single modern human individual had been shown to have a recent Neandertal ancestor in his family tree (*Oase 1*, Fu et al, 2015) This could therefore have been a highly unusual case. Here, we demonstrate that three out of three early individuals from Bacho Kiro Cave had recent Neandertal ancestors [perhaps even four, but due to the lack of data for the individual *CC7-2289* we could not confirm this with certainty]. Therefore, while the interbreeding might not have been ubiquitous, our finds demonstrate that it is likely to have been common among the IUP people to Europe.

Additional points:

The authors refer to the Bacho Kiro remains as the earliest modern humans in the European continent. This should be amended to ‘earliest Upper Paleolithic’ or ‘earliest Late Pleistocene’ modern humans. The discoveries from Apidima cave published recently in this journal suggest that a much earlier modern human dispersal, previously documented from the Near East, also reached Europe.

We have now changed this in line 43 to:

earliest Late Pleistocene modern humans recovered in Europe to date

And in the line 88 to:

the oldest Upper Palaeolithic modern humans in Europe recovered to date¹.

Figure 2 B and C are unclear to non -specialists. What exactly is being plotted here? The legend of the figure should be clarified.

We have now further clarified the legend of the Figure 2B and C.

Referee #5 (Remarks to the Author):

In my opinion reviewer #3’s concerns have been sufficiently addressed.

- Rev. 3 worried of the ground-breaking nature of the data presented in the manuscript to warrant a publication in Nature. The authors (now) provide clear evidence for the ground-

breaking nature of their data warranting publication in Nature.

- o They provide the first genome wide data for the three oldest (circa 45 ky BP) Homo sapiens found in Europe. These three oldest Homo sapiens are also the first ones of that time-period time found in direct association to an archaeological context that can give indications on the cultural identity of these individuals. Before this study, only two European Homo sapiens individuals dated to circa 40,000 years (Oase 1 and Kostenki 14) were (genome-wide) genetically studied and none of them were found in an archaeological context. From their analysis of genome-wide new data, they show that these early modern Europeans “are more closely related to present-day and ancient populations in East Asia and the Americas than to later west Eurasians”. They now observe in the conclusion that the genetic make-up of these earliest European Homo Sapiens from Bulgaria “is consistent with the fact that IUP archaeological assemblages are found from central and eastern Europe to present-day Mongolia, and a putative IUP dispersal that reached from eastern Europe to East Asia”. They refrain from mentioning such an observation in the abstract, and this is legitimate given the work that need to be done to consolidate the knowledge of the IUP from a cultural and genetic point of view.

- o They found that each of the three oldest individuals “had a Neandertal ancestor a few generations back in their family history”. Before only one individual with mixed biological background was known and one could not exclude the possibility that it was a chance find. The analysis of three new individuals out of a total of 5, and 4 of these showing mixed ancestry, is indeed pointing toward an interbreeding pattern with the oldest homo sapiens population in Europe. The abstract terminates with suggesting that “the first modern humans that arrived in Europe mixed frequently with Neandertals”.

- Rev. 3 also worries about “the overinterpretation of genetic data” and the offering “general conclusions about immensely complex topics”.

- o In my opinion, the fact that the authors refrain to refer in the abstract (or even in the title) to the observation made in the conclusion [about the consistency of the genetic signal with “the fact that IUP archaeological assemblages are found from central and eastern Europe to present-day Mongolia, and a putative IUP dispersal that reached from eastern Europe to East Asia”] is showing that they are careful to not overinterpret the data.

- o They study a small number of individuals because it is still the start of such analysis and no other skeletal remains from that time period are available (hence also the ground-breaking nature of the data). Also, the genome of each of these individual is an archive of the genome of (“ten of thousands” say the authors in their rebuttal) of their ancestors. I agree with the authors that “they make no claims to solve hominin demography and dispersal beyond the specific, but central, question discussed in the text about the first modern humans in Europe and whether they interbred with Neandertals” (see authors rebuttal).

About “frequent” interbreeding, it remains unclear to me if the analyzed sampled could be biased against non-hybrids. Could it be that the individuals that entered the archaeological record were preferably of mixed ancestry... because they were less fit than others/because their cadaver were treated/taken care of differently?

We take the point raised by the reviewer about the notion of frequent inbreeding, and have now partially re-written the two sentences in the abstract and in the conclusions section of the paper to read as following:

Lines 51-54

Moreover, we find that all three IUP Bacho Kiro individuals had Neandertal ancestors a few generations back in their family history, confirming that the first European modern humans mixed with Neandertals and suggesting that it could have been common.

Lines 246-250:

Finally, it is striking that all four of the European individuals who overlapped in time with late Neandertals⁷ and from whom genome-wide data have been retrieved had close Neandertal relatives in their family history (Fig. 3, Extended Data Fig. 7 and 8). This suggests that mixing between Neandertals and the first modern humans arriving into Europe was perhaps more common than often assumed.

Human remains from the IUP are extremely rare. Thus, we cannot speculate if “hybrids” were treated differently after death. However, these individuals have Neandertal ancestors 5-7 generations back in their family trees. We thus doubt that they would have been perceived as “hybrids”. But we cannot exclude, of course, that there was some social stigma associated with having some Neandertal ancestry.

It also remains unclear to me where exactly interbreeding happen because only the location of discovery of the analyzed human remains is known. The exact location of the interbreeding event is unknown and could maybe have happen in a different area, maybe even outside of current-days Romania/Bulgaria. Could it have happened outside Europe?

We agree with the reviewer of the notion that we cannot be certain where the interbreeding was taking place. Even though the Bacho Kiro Cave individuals and *Oase 1* had recent Neandertal ancestors in their family trees, the interbreeding of their ancestors could have happen in a different area, or even outside of Europe. Nevertheless, those were different events than the interbreeding that gave rise to the Neandertal ancestry seen in all non-Africans subsequently and today.

Overall, in my opinion the paper warrants publication in Nature. It is a paper that will be of immediate interest to the paleogeneticists, archeogeneticist but also archaeologists and paleoanthropologists. It will likely be abundantly cited.